# Visualizable detection of nanoscale objects using anti-symmetric excitation and non-resonance amplification

Jinlong Zhu [1,2✉], Aditi Udupa [1] & Lynford L. Goddard [1,2]

Why can we not see nanoscale objects under a light microscope? The textbook answers are that their relative signals are weak and their separation is smaller than Abbe's resolution limit. Thus, significant effort has gone into developing ultraviolet imaging, oil and solid immersion objectives, nonlinear methods, fluorescence dyes, evanescent wave tailoring, and point-spread function engineering. In this work, we introduce a new optical sensing framework based on the concepts of electromagnetic canyons and non-resonance amplification, to directly view on a widefield microscope $\lambda/31$-scale (25-nm radius) objects in the near-field region of nanowire-based sensors across a 726-μm × 582-μm field of view. Our work provides a simple but highly efficient framework that can transform conventional diffraction-limited optical microscopes for nanoscale visualization. Given the ubiquity of microscopy and importance of visualizing viruses, molecules, nanoparticles, semiconductor defects, and other nanoscale objects, we believe our proposed framework will impact many science and engineering fields.

[1] Photonic Systems Laboratory, Holonyak Micro and Nanotechnology Laboratory, Department of Electrical and Computer Engineering, University of Illinois at Urbana-Champaign, Urbana, IL 61801, USA. [2] These authors contributed equally: Jinlong Zhu, Lynford L. Goddard. ✉email: zhuwdwz1@illinois.edu

Nanoscale objects that change the steady state of a physical system can be treated as perturbations because they result in an alteration on the function of the host media. Examples include fixed-form nanoscale objects such as semiconductor defects, environmental dust particles, and voids in integrated circuits[1,2], and free-form objects such as dielectric and plasmonic nanoparticles, viruses, and molecule clusters in water and tissue fluids[3,4]. The detection of these nanoscale objects provides a feedback mechanism to control the fabrication quality of functional devices, as well as to understand the physical, biological, or chemical roles performed by the nanoscale objects; thus, their detection is extremely vital to many fields including integrated photonics, quantum chips, human disease diagnosis, medicine, semiconductor circuits, and security systems, to name a few[5–9]. Moreover, there is an increasing interest from industry and academia of not only detecting nanoscale objects, but also visualizing them. We say a method provides visualizable detection if we can look at a captured image with our eyes and differentiate the regions containing an object from regions that contain no objects. This implies that the nanoscale object must produce high contrast compared to the background and to the noise floor of the camera. Visualization therefore enables simple non-contact classification for objects as well as the ability to observe their dynamics in real-time. Typical examples include the visualizable classification of deep subwavelength defects in the semiconductor industry and virus/supermolecule counting for biomedicine[10,11]. Electron microscopy offers nanometric resolution, but it is destructive, slow, costly, and has an ultra-small field-of-view for nanoscale objects. Current optical super-resolution imaging techniques may not be applicable in the visualization of nanoscale objects because fluorescence microscopy requires the object to be fluorescent and scanning near-field optical microscopy is intrinsically slow[10,12–15]. Photothermal imaging has been demonstrated for label-free imaging of individual particles[16,17], but it requires a resonant transition to be pumped. In summary, we need visualizable detection and sensing modalities that meet nearly all the stringent requirements, i.e., label-free, fast, non-destructive, simple in instrumentation, cheap, easy-to-operate, convenient-to-integrate, and large in field-of-view.

Optical far-field-based label-free microscopy may be regarded as the most promising candidate. However, conventionally, using diffraction-limited optical microscopes to detect nanometric objects is extremely challenging not only because of the diffraction barrier, but also the weak Rayleigh scattering[1,18], i.e., a $d^6/\lambda^4$ scaling of the detectable far-field signal is inevitable for a particle with size $d$. This means that a nanometric object can hardly be detected by conventional brightfield or darkfield microscopy because the weak scattering signal results in a low signal-to-noise ratio (SNR). To overcome these issues, interferometric scattering microscopy (iSCAT) positively utilizes the background noise by coherently interfering it with the darkfield scattering signal of the nanoparticles, which boosts the signal of the nanoparticles (scaling with only the 3rd power rather than the 6th power in the Rayleigh scattering regime) because of the interference term[19–21]. Interferometric cross-polarization microscopy (ICPM) works by utilizing the interference of two optical paths with cross-polarization excitation, which enables the background-free detection of nanoparticles[22,23]. A digital microarray can extend both the sensitivity and dynamic range of conventional microarrays by about three orders of magnitude by introducing functionalized gold nanorods as single-molecule labels and an interferometric scanner to rapidly enumerate individual nanorods[24]. These methods have huge potential in the application of detection of isolated free-form nanoscale objects, such as nanoparticles, viruses, and even DNA molecules.

In this paper, we propose a novel visualizable detection modality that uses a simple two-element add-on illumination apparatus and a nanowire assembly, in order to revolutionize conventional low-performance low-numerical-aperture (NA) optical microscopes and enable the detection of multiple nanoscale objects in the near-field region of the nanowire assembly. We demonstrated experimentally that we can visualize the shapes and positions of objects with features as small as a 25-nm radius in a 63-nm wide region (much smaller than the diffraction limit) using the proposed method with 785-nm wavelength light on a low-cost microscope that includes a 0.4 NA objective lens and a low-contrast camera. These experiments demonstrate that fixed-form nanoscale objects can be detected, thus providing an effective solution to the urgent requirement of defect inspection in semiconductor, photonic, and quantum chips.

## Results

**Conceptual features of the framework.** We use a simple artwork in Fig. 1 to briefly present the technical features of our framework and to compare it with other methods. Conventionally, the weak scattering signal (for instance, the intensity, as vividly represented by a crab in Fig. 1a), can be easily overwhelmed by the background signal that is induced by the scattering from surrounding patterns/substrates as well as by fluctuations induced by system errors and the instability of the instruments in a conventional microscope[10]. See the artwork in Fig. 1a. To overcome this, researchers have used resonant sensors (such as ring resonators or metallic particles, which work by observing a spectral shift of the resonance wavelength due to the object[9,25,26]; see Fig. 1b). Although these methods have high sensitivity, they cannot differentiate multiple small objects from a single strong object. Moreover, in a resonator sensor, fluctuations in the ambient environment such as a local temperature change can also produce a resonance shift and thus be confounded with the sensing of an object. Our proposal overcomes the aforementioned limitations of conventional methods by artificially creating an electromagnetic canyon (EC; the region where background electromagnetic field is null), such that the far-field scattering of a nanoscale object amplified by a non-resonance nanostructure ensemble can be directly imaged in a conventional optical microscope. See the artwork in Fig. 1c showing the concept of ECs. Non-resonance amplification is a universal phenomenon of signal amplification that is broadband and thus eliminates the need for precise control of the ambient environment. However, it is only noticeable after the generation of an EC (we will describe this in detail in the following sections). The generation of the EC is built upon the scalar theory that two point sources coherently emitting 180° out of phase can be resolved in a microscope intensity image regardless of the gap size[27]. Although this two-point resolution scalar theory is known, it has not been previously demonstrated experimentally because of the difficulty in creating point sources that emit out of phase. Here, we present the two-beam illumination apparatus as a way to create this anti-symmetric excitation and thus realize the EC. Further, we present the vector theory for two-point resolution in the Supplementary Information. We also introduce mode expansion theory to analytically explain the non-resonance amplification, which is a different phenomenon than out of phase emission.

**Physical models of ECs and non-resonance amplification.** To explain the visualizable non-resonance nanoscale detection, let us consider a pair of parallel nanowires with identical material and topology as shown in Fig. 2a. A scarlet colored cuboid representing a nanoscale object approaches the nanowires and thus perturbs the electromagnetic modes. Here, we do not assume any specific type of material or topology for the object; thus, in practice, it can represent many types of objects including viruses, nanoparticles, supermolecules, environmental dust particles, or even fabrication errors on the nanowires. An eigenmode

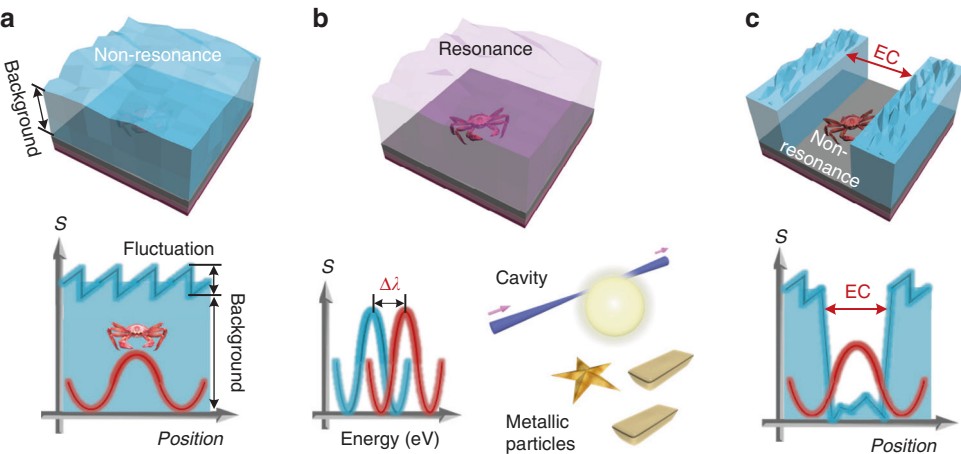

**Fig. 1 Concept artwork explaining visualizable nanoscale detection with an electromagnetic canyon. a** Artwork showing the difficulty in sensing nanoscale objects (represented by a crab) in conventional imaging modalities because of the sea of electromagnetic background signals (represented by blue water) and fluctuations (represented by sawtooth waves). **b** Artwork and schematic showing the mechanisms of resonance-based nanoscale sensing such as the shift in the mode spectrum of a microcavity or of the localized surface plasmon resonance in a metallic particle system. **c** Artwork showing the proposed framework of visualizable nanoscale sensing by artificially creating an electromagnetic canyon. The blue and red curves in each sub-figure denote the electric field strength of the background and of the nanoscale object, respectively. The crab, star, and gold bars were adapted from TurboSquid with the Royalty Free License (all extended uses).

expansion analysis combined with coupled mode perturbation theory shows that the electric field $\mathbf{E}_{SP}(\mathbf{r})$ around the symmetry plane (SP) (see the transparent plane marked with "SP" in Fig. 2a and b) can be represented by (see Supplementary Note 1)[28–31]

$$\mathbf{E}_{SP}(\mathbf{r}) = \mathbf{E}_{SP}\left(\mathbf{r}, c_1^{k_i}\right) + \mathbf{E}_{SP}\left(\mathbf{r}, c_2^{k_i}\right) + \mathbf{E}_{SP}\left(\mathbf{r}, \Delta c_1^{k_i}\big|_{\text{pert}}\right) \\ + \mathbf{E}_{SP}\left(\mathbf{r}, \Delta c_2^{k_i}\big|_{\text{pert}}\right) + \mathbf{E}_{SP}(\mathbf{r}, c_3^u), \quad (1)$$

where $c_1^{k_i}$ and $c_2^{k_i}$ are unperturbed eigenmode expansion coefficients corresponding to the first and second nanowires, respectively. $\Delta c_1^{k_i}\big|_{\text{pert}}$ and $\Delta c_2^{k_i}\big|_{\text{pert}}$ are the perturbation coefficients representing the impact of the object on the first and second nanowires, respectively. $c_3^u$ is the coupling matrix with respect to the object. Hence, $\mathbf{E}_{SP}(\mathbf{r}, c_1^{k_i}) + \mathbf{E}_{SP}(\mathbf{r}, c_2^{k_i})$, $\mathbf{E}_{SP}(\mathbf{r}, \Delta c_1^{k_i}\big|_{\text{pert}}) + \mathbf{E}_{SP}(\mathbf{r}, \Delta c_2^{k_i}\big|_{\text{pert}})$, and $\mathbf{E}_{SP}(\mathbf{r}, c_3^u)$ can be physically interpreted as the field contributions induced by the nanowires, the coupling between the object and the nanowires, and the object itself, respectively. For conventional excitation modes (a single incoherent or coherent excitation beam) in a brightfield or darkfield microscope, the transverse area of the nanowires is much smaller than the beam size. Therefore, the local gradient of the electric field can be neglected and the pair of nanowires are excited analogously and re-radiate in an identical manner on the SP in the near-field region, i.e., $\mathbf{E}_{SP}(\mathbf{r}, c_1^{k_i}) = \mathbf{E}_{SP}(\mathbf{r}, c_2^{k_i})$. A permanent constructive interference is thus formed on the SP. Because the dimensions of the nanowires are much larger than those of the object, the sum of the object-related terms, $\mathbf{E}_{SP}(\mathbf{r}, \Delta c_1^{k_i}\big|_{\text{pert}}) + \mathbf{E}_{SP}(\mathbf{r}, \Delta c_2^{k_i}\big|_{\text{pert}}) + \mathbf{E}_{SP}(\mathbf{r}, c_3^u)$, are overwhelmed by the constructive interference $\mathbf{E}_{SP}(\mathbf{r}, c_1^{k_i}) + \mathbf{E}_{SP}(\mathbf{r}, c_2^{k_i})$ from the background. This is one of the reasons why conventional low-SNR optical microscopes are unable to detect nanoscale objects. However, if we can somehow excite the pair of nanowires into an anti-symmetric state, i.e., $\mathbf{E}_{SP}(\mathbf{r}, c_1^{k_i}) = -\mathbf{E}_{SP}(\mathbf{r}, c_2^{k_i})$, we would create an extended EC on the entire SP because the dominant field $\mathbf{E}_{SP}(\mathbf{r}, c_1^{k_i}) + \mathbf{E}_{SP}(\mathbf{r}, c_2^{k_i})$ would disappear. In this extended EC, only the field contribution from the object-related terms:

$\mathbf{E}_{SP}(\mathbf{r}, \Delta c_1^{k_i}\big|_{\text{pert}}) + \mathbf{E}_{SP}(\mathbf{r}, \Delta c_2^{k_i}\big|_{\text{pert}}) + \mathbf{E}_{SP}(\mathbf{r}, c_3^u)$ would remain. If the object lies exactly in the SP, then the coupling between the object and nanowire would also cancel out, $\mathbf{E}_{SP}(\mathbf{r}, \Delta c_1^{k_i}\big|_{\text{pert on SP}}) = -\mathbf{E}_{SP}(\mathbf{r}, \Delta c_2^{k_i}\big|_{\text{pert on SP}})$ and we would lose the amplification effect; we would be left with only the signal from the isolated object, $\mathbf{E}_{SP}(\mathbf{r}, c_3^u)$. But, if the object does not lie in the SP, we have $\mathbf{E}_{SP}(\mathbf{r}, \Delta c_1^{k_i}\big|_{\text{pert}}) \neq -\mathbf{E}_{SP}(\mathbf{r}, \Delta c_2^{k_i}\big|_{\text{pert}})$ and thus we have amplification provided that:

$$\left| \mathbf{E}_{SP}\left(\mathbf{r}, \Delta c_1^{k_i}\big|_{\text{pert}}\right) + \mathbf{E}_{SP}\left(\mathbf{r}, \Delta c_2^{k_i}\big|_{\text{pert}}\right) + \mathbf{E}_{SP}(\mathbf{r}, c_3^u) \right|^2 \gg \left| \mathbf{E}_{SP}(\mathbf{r}, c_3^u) \right|^2. \quad (2)$$

Therefore, assuming the object is not on the SP, we can engineer the shape and dimensions of the nanowires such that the total object signal is significantly amplified. The aforementioned phenomena for symmetric and anti-symmetric excitation can be understood by analogy as the constructive and destructive interference between two wave trains that move in opposite directions; see the schematics shown in Fig. 2a and b. The detectability of an object relies on the fact that the strength of scattering is larger than the measurement noise and error, $\varepsilon$, of the detection system. For a low-performance detection system or an extremely small object, the uncoupled scattering from the object alone $\left| \mathbf{E}_{SP}(\mathbf{r}, c_3^u) \right|^2$ is usually dominated by $\varepsilon$. However, we can use non-resonance amplification to make the signal detectable, i.e.,

$$\left| \mathbf{E}_{SP}\left(\mathbf{r}, \Delta c_1^{k_i}\big|_{\text{pert}}\right) + \mathbf{E}_{SP}\left(\mathbf{r}, \Delta c_2^{k_i}\big|_{\text{pert}}\right) + \mathbf{E}_{SP}(\mathbf{r}, c_3^u) \right|^2 \gg \varepsilon. \quad (3)$$

The reason we call $\left| \mathbf{E}_{SP}(\mathbf{r}, \Delta c_1^{k_i}\big|_{\text{pert}}) + \mathbf{E}_{SP}(\mathbf{r}, \Delta c_2^{k_i}\big|_{\text{pert}}) + \mathbf{E}_{SP}(\mathbf{r}, c_3^u) \right|^2$ non-resonance amplification is that it is a universal phenomenon that happens at arbitrary wavelengths (see the derivations in the Supplementary Information, in which we did not impose any assumptions on the wavelength), whereas resonator sensors only work around their resonant wavelengths. We should emphasize that the non-resonance amplification also

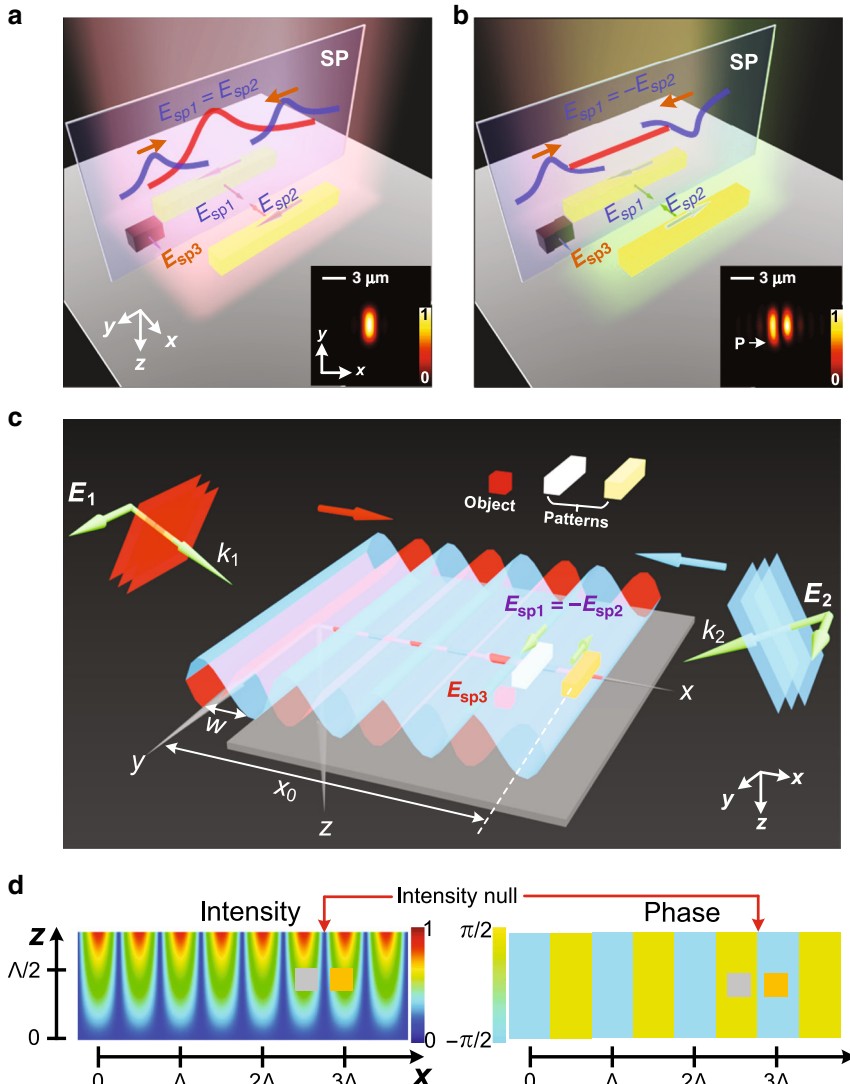

**Fig. 2 Simplified schematics and experimental implementation of the proposed framework. a** and **b** Nanostructure ensemble consisting of a pair of dielectric nanowires and an object on a substrate under **a** conventional and **b** anti-symmetric excitation modes. Each nanowire is 20 nm wide by 800 nm long by 60 nm tall. The edge-to-edge gap for the nanowires is 100 nm. The object is a short nanowire that is 20 nm wide by 100 nm long by 60 nm tall. The distance between the center of the object and symmetry plane (SP) is 30 nm. The insets at the bottom right-hand corner are the corresponding brightfield images. $E_{sp1}$ and $E_{sp2}$ denote the unperturbed electric field on the SP arising from the left and right nanowires. $E_{sp3}$ is the simplified form of the last three terms on the right-hand side of Eq. (1). **c** Generation of the anti-symmetric excitation modes at many different locations for the nanostructure ensemble along the sample using two-wave interference with inclined angles. The different colors (white and yellow) used for the pair of nanowires in **c** indicate that the system is excited into the anti-symmetric mode. **d** The intensity and phase of the standing wave versus $x$ and $z$. The pair of nanowires (represented by a gray and a yellow box) need to be positioned symmetrically with respect to an intensity null such that the anti-symmetric condition $\mathbf{E}_{sp1} = -\mathbf{E}_{sp2}$ holds. The position of the selected intensity null is pointed out by red arrows in **d**.

exists in the case with symmetric excitation $[\mathbf{E}_{SP}(\mathbf{r}, c_1^{k_i}) = \mathbf{E}_{SP}(\mathbf{r}, c_2^{k_i})]$, but as discussed, the background scattering from the nanowires $|\mathbf{E}_{SP}(\mathbf{r}, c_1^{k_i}) + \mathbf{E}_{SP}(\mathbf{r}, c_2^{k_i})|$ is much stronger than that from the object. Hence, the generation of an EC is a prerequisite for the useful application of non-resonance amplification for object detection. The EC generated by the anti-symmetric excitation has an extraordinary feature, i.e., it exists regardless of the gap size between the pair of nanowires. This is easy to understand because by symmetry, the field contributions from the pair of nanowires exactly cancel at the centroid of the nanowire pair. In practice, the nanowires cannot be made arbitrarily small or perfectly identical due to fabrication limitations. However, they can be made quite small and nearly identical. Using 110-oriented crystalline silicon wafers, Iida and

co-workers etched arrays of 12-nm wide lines with 12-nm spacings that were 20 nm tall and also isolated lines with 0.72 nm line edge roughness[32]. Thus, a pair of tightly spaced nanowires can be made identical to a level of less than about 2 nm of variation. Moreover, the surface roughness of the wafer can be easily controlled within 2 nm[32]. This indicates that the background roughness (line edge and wafer surface) is small compared to the 20-nm or larger size of a typical bio-molecule such as a virus. However, because surface roughness is distributed over a large area and may present a strong index contrast, an optimized fabrication process to reduce surface roughness may be needed to visualize individual bio-molecules. The generation of the EC can also be elegantly explained with the dipolar approximation for the pair of nanowires. A systematic study on the EC based on two-dipole interference can be found

in Supplementary Note 2, in which we have analytically and numerically validated various features of ECs. A paradigm-shifting result is that we can realize ECs that are spatially distributed by fabricating an array of nanowire pairs with each pair having a small footprint (size is limited only by the fabrication method).

It is clear now that the key is to generate an EC (via anti-symmetric excitation), such that the non-resonance amplification is detectable. The insets on the bottom right corners of Fig. 2a and b are the brightfield images theoretically predicted by near-field computation and Fourier optics for symmetric (conventional) and anti-symmetric excitations, respectively[33,34]. See Supplementary Note 3 for more details about the simulation methods. Apparently, one can clearly observe an EC between the pair of nanowires and find the object (marked by "P") from the elongated left pattern in the inset of Fig. 2b, whereas one can only find the information of the nanowires in the symmetric excitation case shown in the inset of Fig. 2a.

**Configuration and modeling.** One may ask, how can we excite the pair of nanowires anti-symmetrically especially for a sub-diffraction-limited gap using macroscale illumination? In fact, two-beam interference, which is widely used in optical interferometry provides the answer[35,36]; however, here, we will leverage the phase discontinuity in the standing waves to achieve anti-symmetric excitation and generate a local EC (i.e., an EC that is perfect at the centroid of the nanowires and is weak at other places on the SP). As schematically shown in Fig. 2c, two in-phase $y$-polarized plane waves from distant locations along the $\pm x$-axis impinge at oblique angles on a sample located near the origin. For an explanation of this choice of geometry, see Supplementary Note 2, and in particular Supplementary Table 1, which summarizes field symmetries for different excitation configurations. These two plane waves interfere to produce the standing wave electric field excitation $\mathbf{E}_s = E_0 \cos(2\pi x/\Lambda)\hat{\mathbf{y}}$ with periodic phase jumps (from $+\pi/2$ to $-\pi/2$, as shown in the phase distribution map of Fig. 2d), where $\Lambda = \lambda/\sin\theta$ is the interference period. Here $\theta$ is the incident angle determined as the angle between the incident wavevector and the normal of the wafer surface. Note the period for intensity is $\Lambda/2$. Thus, we can excite the anti-symmetric state by positioning the pair of nanowires at $x = (m + 1/2)\Lambda/2 \pm p$, respectively, where $m$ is any integer and $2p$ is the center-to-center spacing of the nanowires. See the intensity and phase distribution of the standing waves in Fig. 2d, which shows the pair of nanowires being anti-symmetrically excited at a representative position. In other words, for anti-symmetric excitation, the pair of nanowires need to be positioned symmetrically with respect to an intensity null [the one located at $x = (m + 1/2)\Lambda/2$]. A nanoscale object that is close to the anti-symmetrically excited nanowires then can be visualized by a conventional brightfield microscope.

Based on the two-beam anti-symmetric excitation, we first use simulation to investigate the field enhancement of various fixed-form objects on a pair of nanowires (to mimic the defects in a semiconductor integrated circuit chips) and compare the results with those of three conventional methods: darkfield, brightfield, and iSCAT imaging in its initial implementations. Figure 3a shows schematics of the four imaging modalities. Figure 3b shows the optical images for the four imaging modalities for the structure shown in the insets. The structure for the top row of Fig. 3b is an object-free nanowire pair while the bottom row adds a fixed-form end-extrusion object to the bottom nanowire. We can only visualize the nanowire pair and the object through the proposed framework. Because of the nanowires, the scattering signal of the object captured by all the four imaging modalities is

amplified (compared to the case of an isolated object). Thus, the signal will be above the noise floor of the camera in practice. Therefore, visualizability now only depends on whether the contrast is sufficient for our eyes to notice the object. The contrast is defined by contrast $= \frac{S_{per} - S_{mir}}{S_{line}}$, where $S_{per}$ is the peak value of intensity at the position of object in the imaging space, $S_{line}$ is the intensity at the center of the bottom nanowire (the intersection point of the mirror plane and the bottom nanowire), and $S_{mir}$ is the intensity at the mirror position of object with respect to the mirror plane, i.e., where there is no object; see the schematic on the left panel of Fig. 3c. We choose contrast as the figure of merit because it characterizes the distinguishability of an object from the background. Our proposed framework has the highest contrast for all sizes of object. The contrast of brightfield microscopy and iSCAT is very low because the background reflection is much stronger than the scattering signals of nanostructures and its contribution cancels out in the numerator but not in the denominator. The contrast is higher in darkfield microscopy than in brightfield and iSCAT but is lower than in the proposed framework (see Fig. 3c contrast curves). Despite the high contrast, darkfield microscopy is unable to visualize the individual nanowires. ICPM reduces the background compared to iSCAT. The method, which uses an illumination cone and polarizers, is difficult to simulate. However, we can still qualitatively estimate that its contrast should be much better than that of iSCAT but not as good as that of darkfield microscopy. This is because ICPM suppresses the background signal, but only utilizes part of the scattering field of the objects. Moreover, the optical image of ICPM for the pair of nanowires with the object would be similar to that of iSCAT (but without the background). In such an image, it would be difficult to detect the object that is buried in the scattering signal of the nanowires.

We now explore how our proposed method handles a diverse set of fixed-form and free-form objects. Figure 3d presents the optical images of fixed-form objects (parallel bridge, double-sided end extrusion, middle expansion, and side extrusion objects on the nanowires). We can clearly find all these objects in the images. Next, we position a free-form nanoparticle at different locations around the nanowires. Figure 3e shows the computed far-field images. Again, we can clearly observe the positions of the nanoparticles in a sub-diffraction-limited volume. One key exception occurs when the object is located exactly in the middle of the pair of nanowires (see Fig. 3e). In that case, one cannot find the object from the image because the coupling terms cancel out at the object location. This exactly shows the anti-symmetric nature of the EC. Because the SP is only a single line in the sample space and because the shapes of real objects are usually irregular, the probability of failing to see an object is low. Please also see Supplementary Figs. 1–5 for the quantification done on the effects of polarization, phase, numerical aperture, and gap on the visualizability of two point objects. For the task of visualizing an isolated free-form object, our method outperforms brightfield and darkfield because of the non-resonant amplification from the nearby nanowire pair; however, iSCAT and ICPM outperform our method because they do not rely on an EC and thus have a larger signal strength. Advanced iSCAT designs such as inserting a transmissive spatial mask in the back focal plane of a high numerical aperture microscope objective or coating a partially reflective metallic mask onto the center of a glass window[37,38], can significantly enhance the contrast and may even enable routine label-free imaging down to the single-molecule level. In summary, our approach has the main advantage in enabling the visualization of fixed-form objects; it can also enable visualization of free-from objects. Although the signal in our method may not be as strong as that of iSCAT or ICPM, it is adequate for

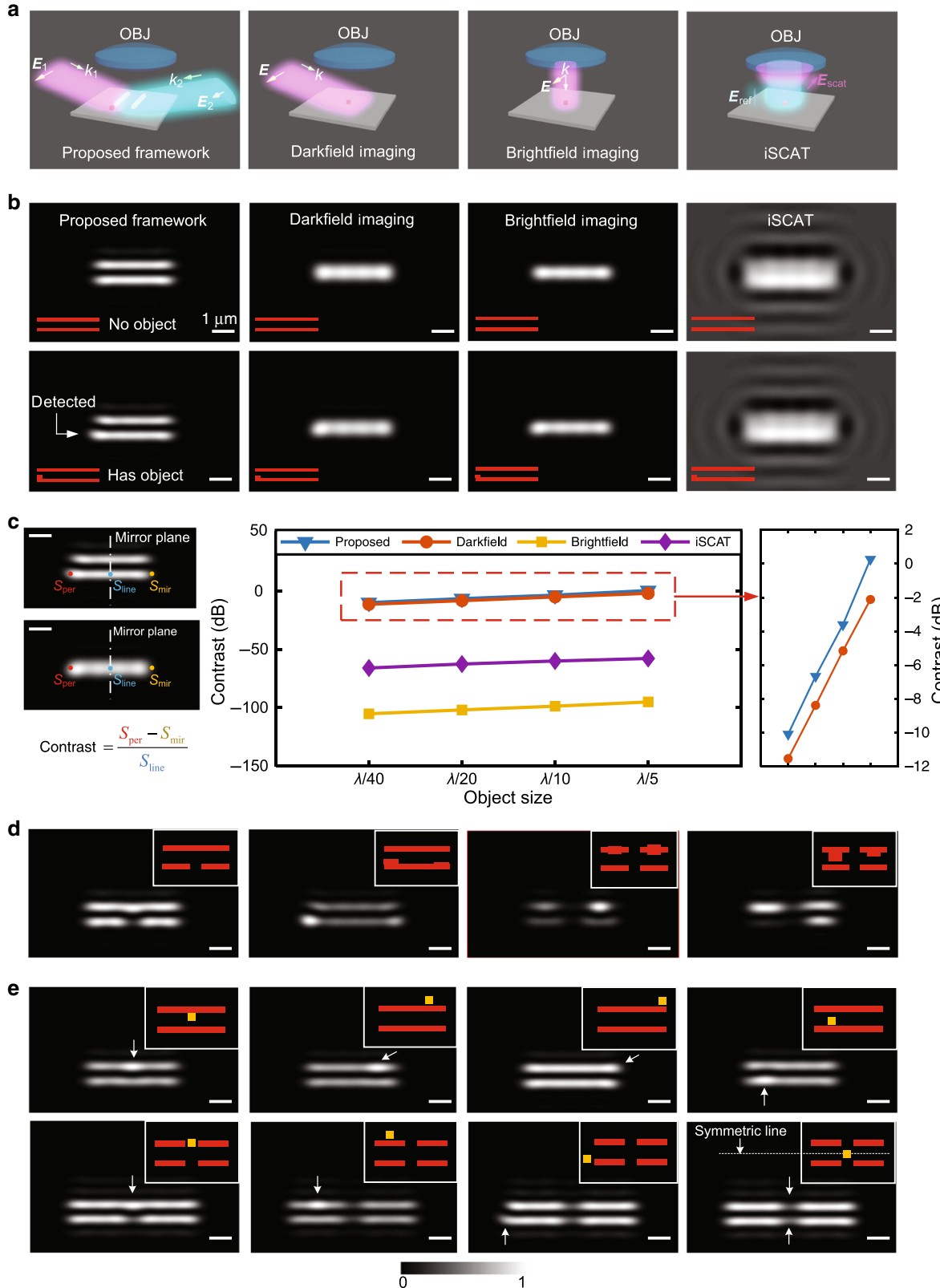

detection because the nanowires can amplify the signal well above the noise floor of a CCD camera.

We should remind our readers not to confuse the proposed framework with structured illumination microscopy (SIM)[39–41]. Whereas SIM works by exploiting interference patterns (moiré patterns) created when two grids are overlaid at an angle, our proposal relies on the localized phase jump in the standing wave, i.e., the two nanowires need to be positioned with respect to an intensity null such that they are excited out-of-phase (see Fig. 2d). If the nanowires are not perfectly positioned but the intensity null is still between the pair, an imperfect EC would be generated. This may still be adequate for visualizing a nanoscale object depending on

**Fig. 3 Simulated performance of different imaging modalities for a diverse set of object types. a** Schematics showing the setups of the proposed framework, darkfield imaging, brightfield imaging, and iSCAT for observing free-form objects. **b** Simulated optical images for unperturbed and perturbed two-nanowire assembly using the proposed framework, darkfield microscopy, brightfield microscopy, and iSCAT, respectively. The individual nanowires and the perpendicular extrusion object can only be noticeably visualized by the proposed framework. **c** Definition of contrast of the object and the investigation of the impact of the dimensions of the object on the contrast for the four imaging modalities. Both the length and the width of the object are scaled whereas the size of the nanowires are kept fixed ($\lambda$/13 wide by 5$\lambda$ long with a $\lambda$/2 side-to-side gap). The right plot is a zoomed-in view of the contrast curves of the proposed framework and darkfield microscopy. **d** Simulated far-field images using the proposed method for fixed-form nanoscale objects on the nanowires. **e** Simulated far-field images using the proposed method for a free-form nanoscale object at different positions around a double-nanowire and a quad-nanowire structure, respectively. The center-to-center spacing of the nanowires for **d** and **e** is 0.4$\lambda$. For the simulation of brightfield images, we only consider the scattering from nanowires and the object by using localized illumination in the simulation, such that the background reflection from the substrate and boundary effect can be removed. The red and yellow colors represent the nanowires and free-form objects, respectively. Figures **b**–**e** have the same scale bar (1 $\mu$m). The input NA of the objective is 0.4 in the simulation in order to match that of the experiments.

its size. A direct evidence is that SIM will result in a resolution improvement not exceeding a factor of two, while the anti-symmetric excitation will result in the generation of an EC regardless of the gap size (see Supplementary Fig. 3a). We should mention that a pair of nanowires is not the exclusive choice, but any other pair of identical nanostructures with arbitrary geometries can be utilized to generate the EC (see Supplementary Note 4 and Supplementary Fig. 8). This opens up a space for engineering the non-resonance sensor such that $|\mathbf{E}_{SP}(\mathbf{r}, \Delta c_1^{k_i}|_{pert}) + \mathbf{E}_{SP}(\mathbf{r}, \Delta c_2^{k_i}|_{pert}) + \mathbf{E}_{SP}(\mathbf{r}, c_3^u)|^2$ can be maximized at a given excitation wavelength. The reason why we choose nanowires is that they can excite strong bright modes when the polarization of the illumination beam is parallel to the long axis of nanowires[42,43].

**Experiments with diverse objects.** We fabricated multiple double-nanowire and quad-nanowire structures (the nominal width is 50 nm) with diverse objects using electron beam lithography (EBL) (see Supplementary Fig. 11 and Supplementary Data 1). Because the nanowire structures are widely used in the semiconductor industry[44–47], we can use them to mimic the detection of objects in typical wafers with isolated patterns[10]. The nanowire structures can also be embedded in liquid environment to sense free-form objects like viruses and molecule clusters, because liquids do not influence the formation of the EC other than to change the interference period $\Lambda$ due to the increased refractive index. We then constructed a two-beam far-field interference system using a single-mode single-frequency 785-nm wavelength laser with integrated optical isolator to implement the proposed idea. We use a top-down microscope with a low numerical aperture objective (0.4 NA) and a low-performance 14-megapixel camera (the dynamic range and SNR of the camera are only 65.3 and 35.5 dB, respectively) to capture 726-$\mu$m × 582-$\mu$m field of view images of the nanowire structures with objects. The schematic and photographs of the detection system are shown in Fig. 4a and Supplementary Fig. 9, respectively. Because the system is operated in the widefield mode, the field-of-view is only limited by the extent of the illumination standing wave and the size of the camera. See a representative full field of view image of the sample captured by the top-down microscope in Fig. 4b. We crop and zoom these images to clearly show selected regions of interest in Figs. 4c–e and 5. Only after completing the optical imaging do we collect scanning electron microscope (SEM) images to corroborate our findings in these regions. The classical Abbe limit of the microscope is 981 nm ($\lambda$/2NA). However, because of the extremely low performance of the imaging system (see more details in Supplementary Note 5), only a 4-$\mu$m gap can be roughly resolved from the brightfield image (see Supplementary Fig. 10b and c); the scattering of an isolated 390 nm × 120 nm nanoscale object can hardly be found from the darkfield image (see Supplementary Fig. 10d). This demonstrates again that we cannot use

conventional brightfield or darkfield imaging modalities to sense nanoscale objects, let alone in cases with fixed-form objects (i.e., defects on the nanowires).

The first step to demonstrate our proposal is to validate the generation of ECs using two-beam interference. Because the upper and lower nanowires need to be positioned at $x = (m + 1/2)\Lambda/2 \pm p$, respectively, we fix the two-beam illumination apparatus and scan the sample along the direction perpendicular to the long axis of nanowires such that the EC with the best contrast can be found visually. Figure 4c shows the SEM image of the fabricated double-nanowire structure that has an 80-nm gap. The sample's orientation is controlled by a rotating stage (RS) to ensure the long axis of the nanowires is parallel to the polarization orientation ($y$-direction) of the field. Figure 4c shows the image of the double-nanowire structure in the best focal plane as the sample is translated along $x$ (i.e., the nanowire centroid is moved relative to the positions of the intensity nulls). The purple ribbon delineates the central region of bright spots from the outside sidelobes. We say that an EC has been generated if there is an intensity minimum at the center of the purple ribbon. Figure 4c shows that the double-nanowire structure undergoes transitions between EC and non-EC in a repeated manner as the sample is moved. The single-nanowire structure, however, does not form an EC regardless of position; Fig. 4d shows that its image always has one bright spot. Figure 4e shows that we can generate ECs for double-nanowire structures with various gaps. As a comparison, we present the darkfield and brightfield images of the same double-nanowire structures captured by the same imaging system; see Fig. 4f and g. Apparently, we cannot visualize the gaps and shapes of the double-nanowire assembly from either of the conventional image sets. The slight fabrication imperfections in some of the nanowires (see the SEM images in Fig. 4e) result in the distortion of patterns in the images of Fig. 4e. This directly demonstrates the sensitivity of our system to nanoscale objects.

We now consider the direct imaging of a diverse set of intentional fixed-form nanoscale objects (defects on the nanowires). Because the objective NA is only 0.4, we choose the 14-$\mu$m long nanowires such that the corresponding optical images look like lines (not dots, like that for 2-$\mu$m long nanowires). This facilitates the observation and classification of the objects. The first sample is a quad-nanowire structure with a tiny dot (represented by "a" in Fig. 5a) positioned in between the upper nanowires, to mimic a typical semiconductor defect or a nanoparticle. See the 3D schematic with the dot marked in red in the left panel of Fig. 5a. An SEM measurement shows that the radius of the dot is around 25 nm. From Fig. 5a, we can clearly and directly observe the perturbing dot in the optical image when the EC is created. When no EC is created, the dot is buried by the background constructive interference. The second sample is a double-nanowire structure with a large object ($\delta_a = 90$ nm, where

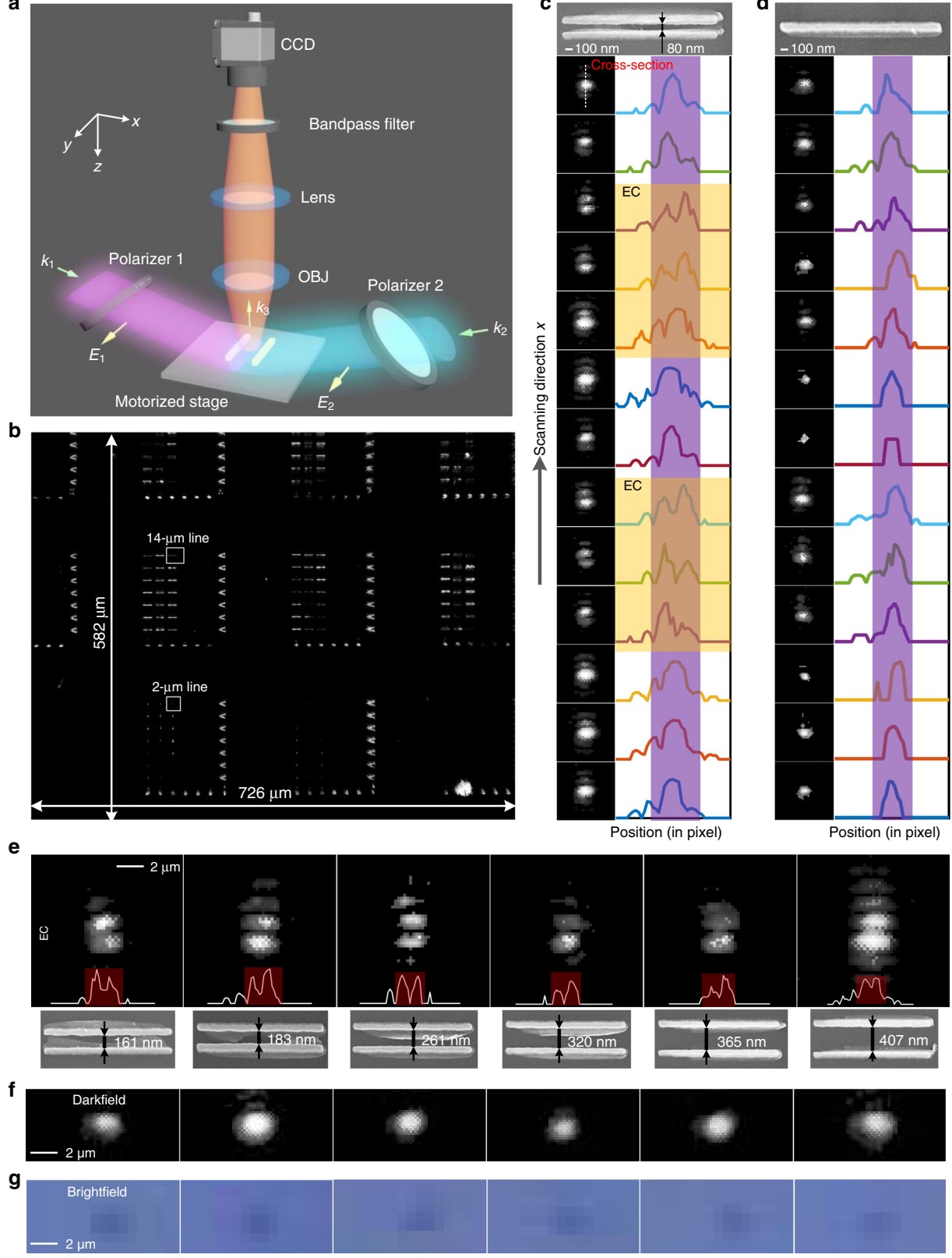

$\delta$ is the width of the object; see the inset in the 3D schematic of Fig. 5b showing the definition of $\delta$) and a small object ($\delta_a = 50$ nm) on the bottom left and bottom right corners, respectively. Moreover, the bottom nanowire is much narrower (32 nm) than the top one. From Fig. 5b optical image with the EC, we can clearly observe a brighter spot in the bottom left corner corresponding to the larger object and that the bottom pattern has a much weaker intensity than that of the top pattern. These details are not visible in the non-EC image. Figure 5c presents a quad-nanowire structure with two central line-expansion objects "a" and "b." The dimension of this type of object can be characterized by the expanded width $\Delta$, defined in the inset in the

**Fig. 4 Experimental setup and results. a** Schematic of the two-beam interference system. **b** A representative widefield image captured by the system of the investigated sample that consists of various two-nanowire and four-nanowire patterns. The two tiny dotted boxes underneath the word "μm" in the labels show typical regions of interest for subsequent parts of Fig. 4. More details of the fabricated patterns can be found in the SEM images shown in Supplementary Fig. 11 and in the pattern layout shown in Supplementary File 1. **c** and **d** Zoomed-in images of the double-nanowire and single-nanowire structures captured at various scan positions of the sample (100-nm increment) along the x-axis. The curves on the right part of **c** and **d** are the corresponding slices along the x-axis. The portions of the curves that are surrounded by a purple ribbon are the main lobes. The curves surrounded by a yellow ribbon correspond to the positions where an EC is formed because anti-symmetric excitation applies. **e** Zoomed-in images of the ECs formed using two-beam interference on double-nanowire structures with various gaps. The bottom line presents the corresponding SEM images of the nanowire assemblies. **f–g** Corresponding darkfield and brightfield zoomed-in images of the double-nanowire structures.

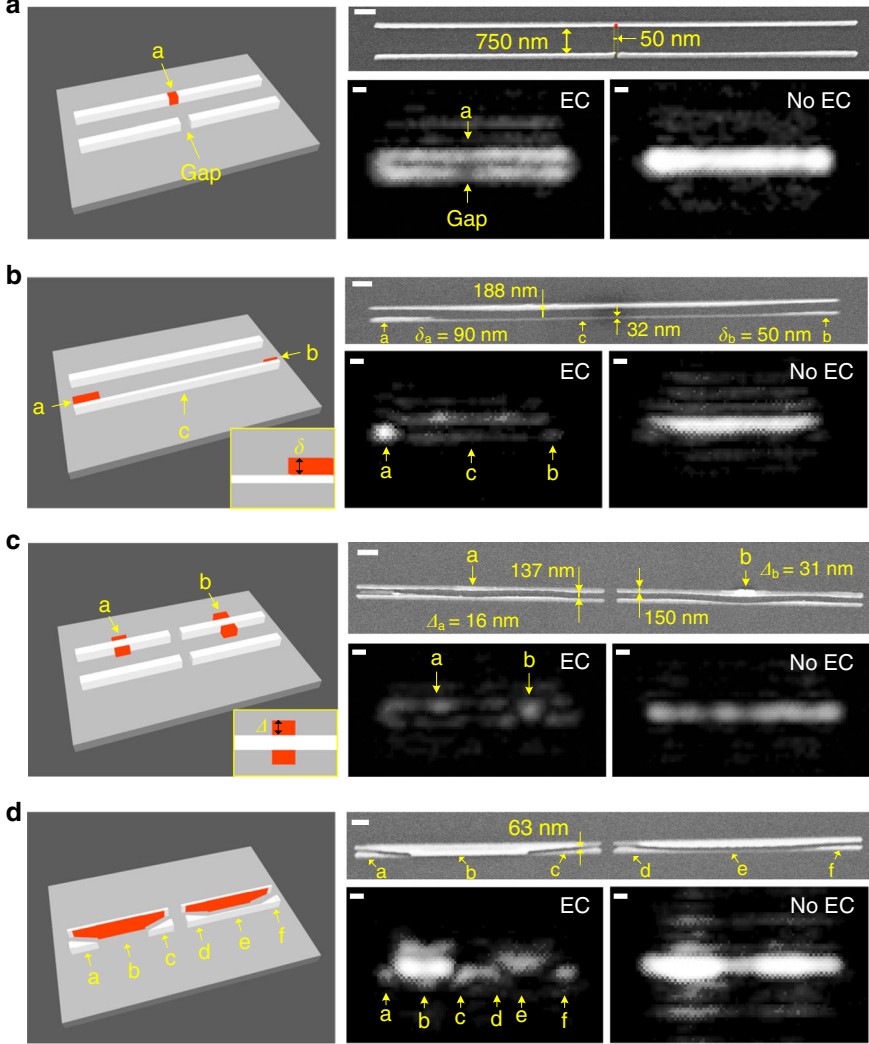

**Fig. 5 Experimental imaging for diverse objects. a–d** Experimental images of patterns with various objects for the anti-symmetrically excited (EC) and symmetrically excited (no EC) cases. All of the gaps in **a–d** are smaller than Abbe's 981-nm resolution limit (actual brightfield resolution is 4 μm because of noise) of the top-down microscope. Objects are marked in red in **a–d**. The scale bars in both the SEM images and the optical images in **a–d** are 500 nm. The nominal width of the nanowires is 50 nm.

3D schematic of Fig. 5c. Similar to those in Fig. 5a and b, one can find two bright spots with the brighter one corresponding to the larger object ($\Delta_b = 31$ nm) in only the EC optical image. A more complicated quad-nanowire structure, shown in Fig. 5d, has tilted 63-nm gaps and two central bumps that break the field symmetry. One can clearly resolve the corners with gaps (marked with "a", "c", "d", and "f" in Fig. 5d) from the EC optical image. Moreover, because the left central bump is thicker than the right one [marked by "b" and "e," respectively], the excited bright modes for the EC optical image are stronger on its left half compared to

on its right half. These images clearly demonstrate that diverse objects, including the isolated, bonded, expanded, and defective ones, can all be detected. One can further validate the effectiveness of the proposed framework by comparing the four experimental images with the simulated images in Fig. 3d, which are computed for the objects that resemble the experimental ones shown in Fig. 5. We should emphasize that all the edge-to-edge gaps (750, 188, 137, and 63 nm) between the upper and lower nanowires are smaller than Abbe's limit of 981 nm. This validates our derivation that an EC can be generated regardless of the gap

size. Thus, the sensing of objects can be implemented in a nanometric region. But we should emphasize that in order to sense even smaller objects (e.g., 20-nm scale viruses or molecule clusters), one may need to select a camera with a large dynamic range and to optimize the shape or materials of the nanowire sensors to further boost the signals of interest.

## Discussion

We have experimentally visualized nanoscale objects by introducing a simple add-on illumination apparatus to create an EC combined with a chip-scale field amplifier consisting of nanowires that amplify the object signals. We have also visualized individual nanowires spaced by $0.10\lambda$ to $0.52\lambda$ (see Fig. 4c and e) and by $0.96\lambda$ (see Fig. 5a). Compared to existing detection modalities, the proposed framework has five major advantages. First, it enables the direct imaging of nanoscale object in a sub-diffraction-limited volume using a conventional microscope. Second, the method uses non-resonance amplifiers to boost the SNR of the object; thus, the prototype can be operated at various wavelengths and without the need for precisely controlling the ambient environment. Third, the nanowire-based amplifier is easy to fabricate using a single EBL and dry-etching step. Fourth, the total device width can be as small as 250 nm (limited only by our EBL) while the length can be selected to best suit the intended application. Fifth, the method uses a widefield microscope and thereby enables arbitrarily large fields of view (limited only by the camera sensor size and the ability to fabricate and align arrays of nanowires to the illumination pattern) to be observed in a single-shot frame. Although our method requires the objects to be in the near-field of the nanowire pair, this limitation can be overcome by using an array of appropriately spaced nanowire pairs, thereby enabling visualization of objects throughout the entire field of view. Unlike other methods, our technique maintains high contrast even in applications where the nanoscale objects are inherently embedded in nanoscale background patterns (e.g., semiconductor defect inspection and quantum chip monitoring)[4].

We believe our work opens up a new avenue for visualizable nanoscale detection and sensing. The technique enables optical microscopy to solve challenging problems across many fields including nanoparticle sensing, biosensing, material characterization, virus counting, and microfluidic monitoring. For example, one can use the developed technique to inspect sub-20 nm defects in semiconductor chips. One may also apply the same prototype for the visualizable sensing of biological objects (e.g., viruses or molecule clusters) by choosing nanowires with optimized geometry and proper refractive index (to make it comparable to that of the biological objects) and patterning functional groups around nanowires (see Supplementary Note 6 and Supplementary Figs. 12 and 13). Once target analytes are trapped, they act as objects that are directly visualized from the optical images. This technique potentially provides a different route to the well-known resonator biosensors for in vitro disease diagnosis, such as the detection of *Coxiella burnetii* in blood plasma from patients[48].

## Methods

**Simulation**. The vectorial images of the electric fields were computed by a three step process. First, each source object was either represented by an electric dipole or was excited using localized plane wave illumination. Next, the near-field was computed using the finite-difference time-domain method and then decomposed into a series of plane waves using far-field projection. Finally, the plane waves within the input NA were propagated through the imaging system using the chirped z-transform. The accuracy of the chirped z-transform was validated against the equivalent magnetic-dipole method in Supplementary Fig. 6. As shown in Supplemental Fig. 7, localized plane wave illumination was used to minimize the effects of the finite computation domain and background reflection that are

prevalent with global illumination. See Supplementary Note 3 for additional details of the simulation methods.

**Experimental setup**. The light source is a fiber-coupled continuous-wave single-mode single-frequency 785-nm wavelength laser (Thorlabs FPV785S) that is split into two output ports using a 50:50 fiber splitter (Thorlabs TN785R5F2). A 1-inch diameter 25-mm focal length lens collimates each output into a free-space beam. Each beam passes through a rotatable free-space linear polarizer (Thorlabs WP25M-UB). Two fiber polarization controllers (Thorlabs FPC020) are inserted in the fiber path between the fiber splitter output ports and the collimating lenses in order to maximize and equalize the powers of the two beams that are transmitted through the free-space polarizers. Two high-precision rotation mounts (Thorlabs PRM1) are utilized to fix the inclined angles of the illumination beams at 60° with a resolution of 5 arcmin for all experiments. The beams then interfere to form the excitation field on the sample. Using a long coherence length laser, matching the optical path lengths, and carefully aligning the angular orientations of the two beams were critical elements for generating ECs across a large area of the sample. See the photograph of the two-beam anti-symmetric excitation apparatus in Supplementary Fig. 9a. By tuning one of the fiber polarization controllers, that beam can be blocked so that the other beam creates conventional darkfield illumination. The sample is mounted on a manual rotation stage (Thorlabs RP01) on top of an *xyz* translation stage (Thorlabs PT3) with motorized actuators (Thorlabs ZST225B) and is scanned relative to the interference field, typically with steps of size 100 nm, to achieve anti-symmetric excitation of a specific nanowire pair. The microscope is a simple 4-*f* imaging system consisting of a Mitutoyo M Plan Apo NIR ×20 objective with 0.4 NA, an ordinary 1-inch diameter 75-mm focal length lens (resulting in ×7.5 magnification), and an Amscope MU1403B 4096 × 3286 pixel CMOS camera with its built-in infrared filter removed to allow the 785-nm light to reach the camera sensor. A 780-nm bandpass filter (Thorlabs FBH780-10) was inserted to reduce the effect of stray room light. A white-light LED (Thorlabs LEDWE-50) is utilized (by the control of a home-made switch box) to image the sample in brightfield mode and locate the patterned areas on the wafer. A full-view photograph of the entire experimental system with the marked optical components in the imaging path is shown in Supplementary Fig. 9c. The entire setup was covered with a plastic cover (not shown) to minimize noise from air drafts in the room.

**System calibration**. The two-beam interference apparatus shown in Supplementary Fig. 9a is aligned by viewing the overlapping shape of the two beams projected on a viewing card until a minimized area of the spot is achieved for the given inclined angle. See the series of photographs showing the alignment process and the beam spot observed in a darkened room in Supplementary Fig. 10a. The entire imaging path is assembled without any image correction algorithms or hardware. In an imaging system, besides the imperfections in optical components, the noise sources (especially the shot noise) in the camera contribute significantly to the distortion, low contrast, and small SNR of captured images. We use a low-cost CMOS camera (MU1403B; see Supplementary Fig. 9b) to validate the robustness of the proposed framework. The observation of structures with deep subwavelength features using conventional imaging modalities is not possible. The imaging performance of the built microscope system is tested with two patterns on a NIST 8820 artifact[49]. Apparently, the images suffer from extremely low contrast and small SNR, and only the 4-μm gap can be barely resolved, although Abbe's limit for the setup is 981 nm (see Supplementary Fig. 9b and c). Moreover, we also cannot detect the information of an isolated nanostructure (390 nm × 120 nm) from the darkfield imaging mode; see the darkfield image and SEM measurement of the nanostructure in Supplementary Fig. 10d.

**Sample fabrication**. Rogue Valley Microdevices fabricated four-inch diameter, 525-μm thick, p-type, Si wafers with a 6-μm-thick wet thermal oxide and 150-nm-thick stoichiometric nitride layer deposited by low-pressure chemical vapor deposition (LPCVD). This layer structure is not important but is presented here for completeness. We cleaved one such wafer into 2.2 cm × 2.2 cm square pieces. Next, we performed EBL using a Raith eLine system set to 10 kV EHT with 30-μm aperture on 300-nm-thick PMMA resist using a 400 μC/cm² dose and 0.2 nA beam current. We deposited 5 nm of Ti and 100 nm of Au, and performed metal liftoff to define single-nanowire, double-nanowire, and quad-nanowire structures with nominal design gaps ranging from 0 to 1000 nm with 50-nm increments. We then deposited a blanket coating of 2 nm of Au to make the top of the wafer electrically conductive. This ensures that we can take SEM images of the exact same sample immediately after optical imaging. See the full SEM view of the entire patterns in Supplementary Fig. 11. The line widths in Supplementary File 1 have a bias of −20 nm on all sides (i.e., 60-nm wide lines were drawn for 100-nm wide designs) to compensate for the finite beam size of the EBL (see Supplementary Note 5). Despite this adjustment, the actual gaps, denoted in the SEM images, are slightly different than the designed dimensions due to fabrication imperfections, including the proximity effect (which was not corrected for). See the SEM images of two representative double-nanowire structures with nanoscale variations in Supplementary Fig. 11b and c.

## Data availability

The data that support the findings of this study are available from the corresponding author upon reasonable request.

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

## Acknowledgements

This work was funded by Cisco Systems Inc. (Gift Awards CG 1141107 and CG 1377144), University of Illinois at Urbana-Champaign College of Engineering Strategic Research Initiative, and Zhejiang University—University of Illinois at Urbana-Champaign (ZJUI) Institute Research Program. The work was primarily carried out in the Holonyak Micro and Nanotechnology Laboratory, University of Illinois; the final EBL patterning was performed in the Materials Research Laboratory Central Research Facilities, University of Illinois. The authors thank Dr. Edmond Chow and Dr. Tao Shang for performing EBL of the initial and final samples, respectively. The authors gratefully acknowledge Cisco System Inc. for access to its Arcetri cluster. L.L.G. acknowledges the Center for Advanced Study at the University of Illinois for teaching release time.

## Author contributions

J.Z. and L.L.G. conceived the concept. J.Z. developed the theory. J.Z. developed the programs and performed the analysis and simulations. A.U. fabricated the samples and collected the SEM images. J.Z. and L.L.G. designed and built the experimental system. J.Z. captured the experimental data. All authors contributed to writing the paper. L.L.G. supervised the project.

## Competing interests

J.Z. and L.L.G. claim a U.S. patent on the presented method in this work through the University of Illinois at Urbana-Champaign.
