## [Peer Review File · Nature Communications]

Peer Review File –

Reviewers' comments Round 1:

Reviewer #1 (Remarks to the Author):

I have read the manuscript on “Visualizable Detection of Nanoscale Perturbations” by Jinlong Zhu et al. with interest. It presents an interesting approach to utilize fabricated device structures itself or an underlying wire-pattern in the sample to be studied to resolve fabrication defects and or attached particles.

Even though, based on the theory and data shown, I’m not at all convinced that the approach would in fact have the potential to be able to detect small perturbations, single viruses or molecular clusters this work would have the potential to stimulate new and interesting lines of thinking. This in itself would for me make this manuscript potentially suitable for publication if the authors are able to convincingly address the following comments.

1) In equation 1 the authors describe a mathematical framework and specify the physical meaning of each of the terms in the equation in the text below. To remove the direct contribution of the nanowires to the observed signal they introduce a phase change of 180 degrees for light exciting one of the nanowires. I however find it very surprising that this same 180 degrees phase change is not applied in the perturbation related terms shown in Equation 2 and the surrounding text describing this (as is also the case for the rest of the manuscript).

To me it seems that a minus sign should also have been included for one of the terms relating to the coupling between the perturbation and the nanowires as effectively these terms correspond to the field that originates from one of the nanowire and is scattered from a perturbation. As a result one of these terms should hence pick up the same 180 degree phase change when the field of one of the nanowires is changed by 180 degrees.

Putting in the appropriate minus sign has the additional benefit that it shows explicitly in the equations that the signal disappears when the perturbation is in the symmetry plane instead of just putting it in words, while at the same time reflecting that the non-resonant enhancement goes down when a perturbation is moved away from one of the nanowires when writing out Eq. 3.

2) The authors currently present this work as if it exists “regardless of the gap size” as stated on page 8 and on other places. This statement as currently presented in this manuscript in my opinion can’t be correct, which I think results from the fact that the authors do not consider the radiation pattern of the original field and how this radiation pattern interacts with the used collection NA.

To explain my reasoning for why I think that this statement can’t be correct. The light scattered from two nanowires will generate a radiation pattern similar to a classical two-slit experiment; in phase the primary lobe is at 0 degrees and collected; which for 180 degrees phase difference between the slits splits into two lobes. As long as both the spacing between the nanowires and the collection NA are small enough indeed no light is collected directly from the nanowires for the out of phase illumination as a result as the two scattering lobes are pushed outside of the collection cone. However this analogy with the classical two-slit experiment hopefully shows that if either the collection NA is chosen too big or the spacing between the nanowires is too large multiple radiation lobes will be detected effectively drowning the signal of interest from the perturbation again as is the case for illumination with a constant phase.

As a result the authors should discuss this aspect of their work with greater care as a proper consideration of the angular radiation patterns should show that there are limitations on this approach that are currently not sufficiently clear from the manuscript and would potentially invalidate some of the claims made.

3) In the conclusion the authors present their method as if it is capable to detect single viruses or molecule clusters; however the manuscript does not consider the impact of the limited dynamic

range of the camera in combination with the fact that local intrinsic sample roughness or fabrication imperfections that are not identical with respect to the plane of symmetry will yield a substantial background signal in this approach in the same way as is the case for the discussed iSCAT approach where this background typically results from the intrinsic roughness of the sample glass.

To be able to resolve single viruses or proteins in iSCAT one needs to do subtraction of this background signal to remove this background that does not yet contain a perturbation in the language of this manuscript (as well as combine signal from multiple pixels to improve the dynamic range).

Although in the technique presented in this manuscript a similar approach of subtracting a sequence of frames that do not yet contain the perturbation could potentially be used the authors should discuss in more detail how this type of background can be overcome to convincingly claim that this approach has the potential to see single viruses or molecular clusters. Please note that great care should be taken in this discussion as based on the current manuscript I don't see how they would be able to see very small perturbations that are already present when this background can't be subtracted, despite this being one of the claims in the manuscript.

4) On page 10 the authors state that the contrast in brightfield and iSCAT is very low, which is correct in their initial implementations. However recent work on iSCAT has shown that this background can in fact be substantially reduced through shaping the reference beam/collection optics, which would be appropriate to mention/discuss seeing that it allowed to detect single proteins indicating an excellent signal to background. The relevant papers I refer to here are: Nano Lett, 2017, 17, 1277-1281 and ACS Photonics, 2017, 4, 211-216 from the Kukura and van Hulst group, respectively.

5) There is complementary work that is less well known than the cited iSCAT work that uses low-NA lenses and a wide field of view for sensing that is in many ways complementary to this work that the authors might wish to give refer to as an example. I'm thinking of the paper "Digital Microarrays: Single-Molecule Readout with Interferometric Detection of Plasmonic Nanorod Labels" from the group of Unlu.

Please note that I do not insist that it is included as a citation, but reading this manuscript I think the authors should be definitely be aware of this paper from the Unlu group.

6) Some typos and other comments I spotted upon reading

- page 7: Equation (2) contains a question-mark, which should be a symbol
- page 9: The angle theta that appears here is never explicitly defined and also does not appear in the figure.
- page 10: There is a double full-stop on line 237 after perturbation.
- page 13: Line 308 mentions that the collection NA is 0.4, yet in the SI an NA of 0.8 is used for simulations. I initially was confused that the 0.8 only applies to the simulations, which could have been presented more clearly.

Reviewer #2 (Remarks to the Author):

This is a very interesting paper describing what seems to be a new approach to optical imaging. Once this work is published, the hope is that the microscopy community show the same interest as shown imaging using plasmonic particles. Although the authors are aware:: See for example in the "Optical Imaging of Individual Plasmonic Nanoparticles in Biological Samples" paper in Annual Reviews : <https://www.annualreviews.org/doi/abs/10.1146/annurev-anchem-071213-020125>

The manuscript could be enhanced by providing more detail about how precisely the two nanowires need to be positioned. The manuscript states: "symmetrically with respect to an intensity null such that they are excited out-of-phase".

The figure caption for Figure 4 could be improved by a better description of 4B. It is difficult to see the details in 4E.

Defects on nanowires are nice, but the paper does not present an example that would result in a greater audience such as an improved image of a biological sample.

Reviewer #3 (Remarks to the Author):

The authors present a novel way of detecting nanometer-large objects with a conventional (diffraction-limited) wide-field microscope. To achieve this, they implement near-field effects within two nearby emitters such as two parallel aligned nanowires. This is a clever and nice idea, and the authors achieved to explain and present it in a very nice way. I would really like to see this published.

Unfortunately, I think the authors oversell their story a little bit and some issues still need clarification.

Possible applications such as diagnostics should be discussed in more detail in the introduction and conclusions – gives stronger motivation.

The notation “perturbation” is in my opinion irritating and not well known to the microscope community. What the authors really detect are nanoscale objects.

As the authors highlight (only once) throughout the manuscript, this is a near-field approach and this needs to be stressed more prominently, already starting in the abstract and introduction.

The authors only show the detection of individual perturbations near the nanowires. Resolution is however about distinguishing multiple nearby features, objects or perturbations. Just observing individual perturbations such as individual bacterium or virus is also possible with conventional iSCAT.

The comparison in figure 3 is somehow unfair. Dark-field/bright-field and iSCAT do not need the two parallel nanowires and would be able to detect single perturbations in the absence of the nanowires. Therefore, this might bring up the question why one should use the current approach (despite its simplicity).

Minor issues:

- Page 5, first sentence last paragraph of introduction: “We have” sounds like the authors refer to experiments published in the past.
- Page 8, bottom: Better to use “theoretically predicted” instead of “obtained” to highlight that these are not experimental data (I initially thought so).
- Page 10/11: The description of the comparison to dark-field is somehow missing (or did I miss this)?
- Page 11, bottom: Cite Fig S1-S4 for the quantification done e.g. on the contrast and gap dependence already here.
- Page 13, top: Is the polarization of the laser light important?
- Page 13, bottom: Was the best contrast determined visually or somehow quantitatively?
- Page 14, top: What is “best-focal”?

UNIVERSITY OF ILLINOIS
AT URBANA - CHAMPAIGN

Micro and Nanotechnology Laboratory

Dr. Jinlong Zhu
2231 Micro and Nanotechnology Laboratory
208 North Wright Street
Urbana, IL 61801-2355

We thank the reviewers for the insightful comments, which we have addressed in the revised version. Our responses to reviewers' individual points are shown below in blue and edits to the manuscript are shown in red.

Reviewer #1:

1.1) I have read the manuscript on “Visualizable Detection of Nanoscale Perturbations” by Jinlong Zhu et al. with interest. It presents an interesting approach to utilize fabricated device structures itself or an underlying wire-pattern in the sample to be studied to resolve fabrication defects and or attached particles.

We thank the reviewer for their interest in our paper.

1.2) In equation 1 the authors describe a mathematical framework and specify the physical meaning of each of the terms in the equation in the text below. To remove the direct contribution of the nanowires to the observed signal they introduce a phase change of 180 degrees for light exciting one of the nanowires. I however find it very surprising that this same 180 degrees phase change is not applied in the perturbation related terms shown in Equation 2 and the surrounding text describing this (as is also the case for the rest of the manuscript).

To me it seems that a minus sign should also have been included for one of the terms relating to the coupling between the perturbation and the nanowires as effectively these terms correspond to the field that originates from one of the nanowire and is scattered from a perturbation. As a result one of these terms should hence pick up the same 180 degree phase change when the field if one of the nanowires is changed by 180 degrees.

Putting in the appropriate minus sign has the additional benefit that it shows explicitly in the equations that the signal disappears when the perturbation is in the symmetry plane instead of just putting it in words, while at the same time reflecting that the non-resonant enhancement goes down when a perturbation is moved away from one of the nanowires when writing out Eq. 3.

We thank the reviewer for pointing this out.

This is a point that we need to clarify. Note that the presented equations are vector equations, thus the change in sign due to the change in excitation is already included in the field expression of perturbation terms $\mathbf{E}_{\text{SP}}(\mathbf{r}, \Delta c_1^{k_i} |_{\text{pert}})$ and $\mathbf{E}_{\text{SP}}(\mathbf{r}, \Delta c_2^{k_i} |_{\text{pert}})$. Thus, the 180 degree phase change for the nanowires is reflected in the statement

$\mathbf{E}_{\text{SP}}(\mathbf{r}, \Delta c_1^{k_i} \Big|_{\text{pert on SP}}) = -\mathbf{E}_{\text{SP}}(\mathbf{r}, \Delta c_2^{k_i} \Big|_{\text{pert on SP}})$. If the object lies on the symmetry plane, one can immediately derive that the coupling term cancels out because of $\mathbf{E}_{\text{SP}}(\mathbf{r}, \Delta c_1^{k_i} \Big|_{\text{pert on SP}}) + \mathbf{E}_{\text{SP}}(\mathbf{r}, \Delta c_2^{k_i} \Big|_{\text{pert on SP}}) = 0$. To make this more clearly, we have revised the statement relating to Equation (2). Please see “In this extended EC, only the field contribution from the perturbation-related terms: $\mathbf{E}_{\text{SP}}(\mathbf{r}, \Delta c_1^{k_i} \Big|_{\text{pert}}) + \mathbf{E}_{\text{SP}}(\mathbf{r}, \Delta c_2^{k_i} \Big|_{\text{pert}}) + \mathbf{E}_{\text{SP}}(\mathbf{r}, c_3'')$ would remain. If the object lies exactly in the SP, then the coupling between the object and nanowire would also cancel out, $\mathbf{E}_{\text{SP}}(\mathbf{r}, \Delta c_1^{k_i} \Big|_{\text{pert on SP}}) = -\mathbf{E}_{\text{SP}}(\mathbf{r}, \Delta c_2^{k_i} \Big|_{\text{pert on SP}})$ and we would lose the amplification effect; we would be left with only the signal from the isolated object, $\mathbf{E}_{\text{SP}}(\mathbf{r}, c_3'')$. But, if the object does not lie in the SP, we have $\mathbf{E}_{\text{SP}}(\mathbf{r}, \Delta c_1^{k_i} \Big|_{\text{pert}}) \neq -\mathbf{E}_{\text{SP}}(\mathbf{r}, \Delta c_2^{k_i} \Big|_{\text{pert}})$ and thus we have amplification provided that:

$$\left| \mathbf{E}_{\text{SP}}(\mathbf{r}, \Delta c_1^{k_i} \Big|_{\text{pert}}) + \mathbf{E}_{\text{SP}}(\mathbf{r}, \Delta c_2^{k_i} \Big|_{\text{pert}}) + \mathbf{E}_{\text{SP}}(\mathbf{r}, c_3'') \right|^2 \neq \left| \mathbf{E}_{\text{SP}}(\mathbf{r}, c_3'') \right|^2 . \quad (2)$$

Therefore, assuming the object is not on the SP, we can engineer the shape and dimensions of the nanowires such that the total object signal is significantly amplified” in the revised manuscript.

We hope our clarification addresses the reviewer’s concern.

1.3) The authors currently present this work as if it exists “regardless of the gap size” as stated on page 8 and on other places. This statement as currently presented in this manuscript in my opinion can’t be correct, which I think results from the fact that the authors do not consider the radiation pattern of the original field and how this radiation pattern interacts with the used collection NA.

To explain my reasoning for why I think that this statement can’t be correct. The light scattered from two nanowires will generate a radiation pattern similar to a classical two-slit experiment; in phase the primary lobe is at 0 degrees and collected; which for 180 degrees phase difference between the slits splits into two lobes. As long as both the spacing between the nanowires and the collection NA are small enough indeed no light is collected directly from the nanowires for the out of phase illumination as a result as the two scattering lobes are pushed outside of the collection cone. However this analogy with

the classical two-slit experiment hopefully shows that if either the collection NA is chosen too big or the spacing between the nanowires is too large multiple radiation lobes will be detected effectively drowning the signal of interest from the perturbation again as is the case for illumination with a constant phase.

As a result the authors should discuss this aspect of their work with greater care as a proper consideration of the angular radiation patterns should show that there are limitations on this approach that are currently not sufficiently clear from the manuscript and would potentially invalidate some of the claims made.

We respectfully disagree, but thank the reviewer for these comments and the specific thought experiment as they help us to clarify our statements.

Our arguments for EC formation are based purely on symmetry properties. The fields generated by the nanowires have a closed form analytic expression (see Eq. S11 and S12 in the supplement). For y-polarized dipoles spaced in x and excited into the antisymmetric state, the x , y , and z components of these fields are antisymmetric with respect to the planes $y=0$, $x=0$, and both $x=0$ and $y=0$, respectively. See Table 1 in the supplement. Therefore, we have the following result that was stated in section “2. The role of symmetry in generating an electromagnetic canyon: dipolar approximation” in the supplement: “Anti-symmetry across at least one plane for each near-field component produces a perfect destructive interference in the microscope image at the intersection of the two planes. Thus, the microscope image will have zero intensity at the center point $(x, y) = (0, 0)$.” There is one clarifying assumption that we have added later in that paragraph to address the reviewer’s concern, “Note that we assumed that the centroid of the nanowire pair is located on the optical axis of the microscope so that the system respects field symmetries. Under this assumption, a perfect EC is formed, regardless of the NA of the objective or the spacing of the nanowires. This is not a limiting assumption because in practice, even when the nanowire centroid is off-axis, the spacing is small enough that the radiation pattern from each nanowire is equally collected by the objective.”

To further address the reviewer’s comment, we show that the EC for anti-symmetric excitation always exists regardless of how many scattering lobes (or in other words, diffraction orders) are detected by an imaging system. Let’s first use the two-slit experiment to numerically demonstrate. As shown in Fig (A) and (B), the top and bottom slits (with a side-to-side gap $\lambda/40$) are excited into phase 0 and 180, respectively, and the resulting scattering field is captured by an imaging system with a collection NA = 1.0 (such that all the diffraction orders by the slits can be detected). Because we consider only coherent imaging in linear optics, the image of the two slits can be regarded as the coherent superposition of the images of individual slits. The blue/yellow curve in Figure (C) is the cross-section along the central line of the image pattern (see the inset in the top right corner of (C)) of slit 1/2. The plot on the right-hand side of Fig. (C) is the zoom-in plot of the sidelobes in the image plane. Now let’s look at the phase distribution that corresponds to the amplitude curves in Fig. (D). Apparently, not only are the central lobes of curves 1 and 2 out-of-phase, but the sidelobes of curves 1 and 2 are also approximately out-of-phase. This indicates that destructive interference (at least, partially) also exists between sidelobes of slits 1 and 2 in the image plane (please see Supplementary Figure 1 in the Supplementary Information about the destructive interference for cases of not-so-perfect out-of-phase excitation). Thus, the coherent superposition of the two complex images will result in an electromagnetic canyon in the center, regardless of how many diffraction orders in the object space can be detected by the imaging system.

In order to better help the reviewer and readers to understand this point, we used a general model, i.e., a pair of electric dipoles, to semi-analytically demonstrate why there is a permanent electromagnetic canyon in the middle of the dipoles in the image plane, regardless of the NA. Please see the supplemental section “The role of symmetry in generating an electromagnetic canyon: dipolar approximation” along with Supplementary Figures 1-4 (especially Supplementary Figure 1, in which we show that electromagnetic canyon is generated by a pair of dipoles with the gap size $\lambda/256$ for a high NA 0.8) in the Supporting Information. Any pair of identical nanostructures can be approximately represented by a

pair of electric dipoles (or dipole assemblies); thus, the dipole approximation analysis in the Supporting Information provides a general picture to understand the generation of electromagnetic canyon.

1.4) In the conclusion the authors present their method as if it is capable to detect single viruses or molecule clusters; however the manuscript does not consider the impact of the limited dynamic range of the camera in combination with the fact that local intrinsic sample roughness or fabrication imperfections that are not identical with respect to the plane of symmetry will yield a substantial background signal in this approach in the same way as is the case for the discussed iSCAT approach where this background typically results from the intrinsic roughness of the sample glass.

To be able to resolve single viruses or proteins in iSCAT one needs to do subtraction of this background signal to remove this background that does not yet contain a perturbation in the language of this manuscript (as well as combine signal from multiple pixels to improve the dynamic range).

Although in the technique presented in this manuscript a similar approach of subtracting a sequence of frames that do not yet contain the perturbation could potentially be used the authors should discuss in more detail how this type of background can be overcome to convincingly claim that this approach has the potential to see single viruses or molecular clusters. Please note that great care should be taken in this discussion as based on the current manuscript I don't see how they would be able to see very small perturbations that are already present when this background can't be subtracted, despite this being one of the claims in the manuscript.

Thanks for the insightful comments.

Question 1) Impact of the limited dynamic range of the camera

As the reviewer correctly points out, the small dynamic range of a camera may hinder the sensing of a single virus (or molecule clusters) if the scattering signal of the single virus ($|\mathbf{E}_{\text{SP}}(\mathbf{r}, c_3'')|^2$) is smaller than the noise floor ϵ of the camera (please see “This implies that the nanoscale object must produce high contrast compared to the background and to the noise floor of the camera” on Page 2 in the manuscript). However, using the proposed anti-symmetric excitation, the scattering signal can be boosted to $|\mathbf{E}_{\text{SP}}(\mathbf{r}, \Delta c_1^{k_i}|_{\text{pert}}) + \mathbf{E}_{\text{SP}}(\mathbf{r}, \Delta c_2^{k_i}|_{\text{pert}}) + \mathbf{E}_{\text{SP}}(\mathbf{r}, c_3'')|^2$ such that it goes beyond the noise floor ϵ . Moreover, most viruses vary in diameter from 20 nm to 250–400 nm (please see the webpage <https://www.britannica.com/science/virus/Size-and-shape>), which are close to or even larger than the sizes of some patterned nanoscale perturbations shown in Figure 5 (please see marked sizes of the perturbations in the SEM images of Figure 5). Therefore, we believe we can use the same system to sense single viruses.

To sense smaller objects, as the reviewer pointed out, we do need a camera with a better dynamic range. Therefore, we have revised our manuscript by adding some discussions on the dynamic range of cameras. Please see “We use a top-down microscope with a low numerical aperture objective (0.4 NA) and a low-performance 14-megapixel camera (the dynamic range and SNR of the camera are only 65.3 dB and 35.5 dB, respectively) to capture...” on Page 14. We also added a sentence at the end of Section 2.3 to inform our readers about the importance of the dynamic range of cameras and the engineering optimization for the nanowire sensors in sensing very small objects. Please see “But we should emphasize that in order to sense even smaller objects (for example, 20-nm scale viruses or molecule clusters), one may need to select a camera with a larger dynamic range (than the one we used in our experiments) or to optimize the shape or materials of the nanowire sensors to further boost the signals of interest.”

Question 2) Impact of local intrinsic sample roughness or fabrication imperfections

As the reviewer correctly implied, the nanowires cannot be made arbitrarily small or perfectly identical due to fabrication limitations. However, they can be made quite small and nearly identical.

Using 110-oriented crystalline silicon wafers, Iida and co-workers etched arrays of 12-nm wide lines with 12-nm spacings that were 20 nm tall and also isolated lines with 0.72 nm line edge roughness (see the paper we cited in the manuscript *Development of Standard Samples with Programmed Defects for Evaluation of Pattern Inspection Tools. Proc. SPIE 2019, 10959, 109590J. <https://doi.org/10.1117/12.2514897>*). Thus, a pair of tightly spaced nanowires can be made identical to a level of less than about 2 nm of variation. Moreover, different from the case where a sample glass with an intrinsic large surface roughness being used in iSCAT, in our technique, we use a semiconductor wafer, whose surface roughness can be easily controlled within 2 nm (see the paper https://tsapps.nist.gov/publication/get_pdf.cfm?pub_id=905034). Therefore, nearly identical nanowires (<2 nm line edge roughness) can be fabricated on nearly uniform wafer surface (< 2-nm surface roughness) using modern nanofabrication techniques. This is adequate for us to sense a single virus (> 20 nm) or even molecule clusters, provided an appropriate camera is selected. Therefore, in the presented technique, we do not need the subtraction of a reference frame that does not contain the perturbations in order to eliminate the background signals. Please see the added sentences “In practice, the nanowires cannot be made arbitrarily small or perfectly identical due to fabrication limitations. However, they can be made quite small and nearly identical. Using 110-oriented crystalline silicon wafers, Iida and co-workers etched arrays of 12-nm wide lines with 12-nm spacings that were 20 nm tall and also isolated lines with 0.72 nm line edge roughness.³² Thus, a pair of tightly spaced nanowires can be made identical to a level of less than about 2 nm of variation. Moreover, the surface roughness of the wafer can be easily controlled within 2 nm.³² This indicates that the background roughness (line edge roughness and wafer surface roughness) has limited impact on the boosted scattering signal $\left| \mathbf{E}_{\text{SP}}(\mathbf{r}, \Delta c_1^{k_i} |_{\text{pert}}) + \mathbf{E}_{\text{SP}}(\mathbf{r}, \Delta c_2^{k_i} |_{\text{pert}}) + \mathbf{E}_{\text{SP}}(\mathbf{r}, c_3^u) \right|^2$ of typical objects such as viruses, which usually have size larger than 20 nm in general” on Pages 8 and 9.

1.5) On page 10 the authors state that the contrast in brightfield and iSCAT is very low, which is correct in their initial implementations. However recent work on iSCAT has shown that this background can in fact be substantially reduced through shaping the reference beam/collection optics, which would be appropriate to mention/discuss seeing that it allowed to detect single proteins indicating an excellent signal to background. The relevant papers I refer to here are: Nano Lett, 2017, 17, 1277-1281 and ACS Photonics, 2017, 4, 211-216 from the Kukura and van Hulst group, respectively.

We thank the reviewer for providing such important references.

We have added discussions for these new implementations of iSCAT in the manuscript. Please see “...however, iSCAT and ICPM outperform our method because they do not rely on an EC and thus have a larger signal strength. Moreover, by introducing advanced designs such as inserting a transmissive spatial mask placed into the back focal plane of a high numerical aperture microscope objective or coating a partially reflective metallic mask onto the center of a glass window,^{37,38} the contrast can be significantly enhanced when compared with iSCAT in its initial implementations. These enhanced iSCAT may even enable routine label-free imaging down to the single-molecule level” on Page 12.

1.6) There is complementary work that is less well known than the cited iSCAT work that uses low-NA lenses and a wide field of view for sensing that is in many ways complementary to this work that the authors might wish to give refer to as an example. I’m thinking of the paper “Digital Microarrays: Single-Molecule Readout with Interferometric Detection of Plasmonic Nanorod Labels” from the group of Unlu.

Please note that I do not insist that it is included as a citation, but reading this manuscript I think the authors should be definitely be aware of this paper from the Unlu group.

We thank the reviewer for providing this interesting paper, which helps us to better understand the area of biosensing.

We decided to cite this paper. Please see “A digital microarray can extend both the sensitivity and dynamic range of conventional microarrays by about 3 orders of magnitude by introducing functionalized gold nanorods as single-molecule labels and an interferometric scanner to rapidly enumerate individual nanorods.²⁴” on Page 3.

1.7) Some typos and other comments I spotted upon reading

- page 7: Equation (2) contains a question-mark, which should be a symbol
- page 9: The angle theta that appears here is never explicitly defined and also does not appear in the figure.
- page 10: There is a double full-stop on line 237 after perturbation.
- page 13: Line 308 mentions that the collection NA is 0.4, yet in the SI an NA of 0.8 is used for simulations. I initially was confused that the 0.8 only applies to the simulations, which could have been presented more clearly.

We thank the reviewer for the careful reading.

-page 7. This is a misprint due to a version mismatch either in MathType or Word.

-page 9. We have given a definition for theta. Please see “Here θ is the incident angle determined as the angle between the incident wavevector and the normal of the wafer surface” on Pages 9 and 10.

-page 10. Revised.

-page 13. In the manuscript, both the simulation in Figure 3 and the experiment in Figures 4 and 5 were conducted with a 0.4 input NA of the objective. Please see the added sentence at the end of the caption of Figure 3 “The input NA of the objective is 0.4 in the simulation in order to match that of the experiments.” The reason why we choose 0.8 in the SI is to imply the fact that our technique, in theory, is independent of the input NA due to symmetry. We are sorry for the confusion caused by the lack of more details. Therefore, we have added a sentence at the end of the first paragraph of Section 3 in the SI, see “Here we should mention that any other input NA (for example, NA = 0.4 in the manuscript) can be chosen for the following analysis without affecting the anti-symmetric conclusions.”

Reviewer #2:

2.1) This is a very interesting paper describing what seems to be a new approach to optical imaging. Once this work is published, the hope is that the microscopy community show the same interest as shown imaging using plasmonic particles. Although the authors are aware:: See for example in the "Optical Imaging of Individual Plasmonic Nanoparticles in Biological Samples" paper in Annual Reviews : <https://www.annualreviews.org/doi/abs/10.1146/annurev-anchem-071213-020125>

We thank the reviewer for their interest in our paper as well as the suggestion on increasing the visibility of our paper by providing this helpful review paper.

We have cited this paper in our manuscript as Ref. 4. Please see "...and free-form objects such as dielectric and plasmonic nanoparticles..." in Lines 4 and 5 of Introduction.

2.2) The manuscript could be enhanced by providing more detail about how precisely the two nanowires need to be positioned. The manuscript states: "symmetrically with respect to an intensity null such that they are excited out-of-phase".

We are sorry for the lack of enough details.

We have added more details accordingly. Please see "...the two nanowires need to be positioned with respect to an intensity null such that they are excited out-of-phase; see Fig. 2d showing the positions of two nanowires (represented by a gray square and an orange one, respectively) with respect to an intensity null (pointed out by red arrows). If the nanowires are not perfectly positioned but the intensity null is still between the pair, an imperfect EC would be generated. This may still be adequate for visualizing a nanoscale object depending on its size" at the beginning of Page 13.

2.3) The figure caption for Figure 4 could be improved by a better description of 4B. It is difficult to see the details in 4E.

Great suggestion.

We have revised the caption of Figure 4b by linking it to Supplementary Figure 11 in the Supplementary Information that discloses more geometrical details of the fabricated patterns, in order to meet the requirement of maximum word count. See "A representative wide-field image captured by the system of the investigated sample that consists of various two-nanowire and four-nanowire patterns. The two tiny dotted boxes underneath the word "μm" in the labels show typical regions of interest for subsequent parts of Fig. 4. More details of the fabricated patterns can be found in the GDS file and SEM images shown in Supplementary Figure 11 in the Supplementary Information" in the caption of Figure 4B. We also adjusted the positions of the SEM images such that more details are clearly visible. See the revised Figure 4 in the manuscript.

2.4) Defects on nanowires are nice, but the paper does not present an example that would result in a greater audience such as an improved image of a biological sample.

As the reviewer correctly pointed out, presenting a decent example that shows the sensing of biological samples would make our paper more visible to the community of biomedical science, but due to the fact we work more in the optics and semiconductor materials fields, we would like to focus on the general concept and instrumentation of our technique and use the sensing of nanowire defects as an example to demonstrate the effectiveness of our technique. Therefore, we would like to collaborate with other groups and conduct the sensing of biological samples such as viruses and even single molecules in the future as a complement to demonstrate the generality of our technique.

Reviewer #3:

2.1) The authors present a novel way of detecting nanometer-large objects with a conventional

(diffraction-limited) wide-field microscope. To achieve this, they implement near-field effects within two nearby emitters such as two parallel aligned nanowires. This is a clever and nice idea, and the authors achieved to explain and present it in a very nice way. I would really like to see this published.

We thank the reviewer for their interest in our paper.

2.2) Unfortunately, I think the authors oversell their story a little bit and some issues still need clarification. Possible applications such as diagnostics should be discussed in more detail in the introduction and conclusions – gives stronger motivation.

Thanks for the great suggestion.

We have added more discussions in both the introduction and conclusions accordingly. Please see “These experiments demonstrate that fixed-form nanoscale objects can be detected, thus providing an effective solution to the urgent requirement of defect inspection in semiconductor, photonic, and quantum chips. Moreover, because most viruses (or molecule clusters) vary in diameter from 20 nm to 250-400 nm, we expect that our method can be adapted to sense these biological objects in a manner that is different from the well-known resonator biosensors,¹⁷ thus providing a potential candidate for in vitro disease diagnosis” at the end of Introduction, and “One may also apply the same prototype for the visualizable sensing of biological objects (e.g., viruses or molecule clusters) by patterning functional groups around nanowires. Once target analytes are trapped, they act as objects that are directly visualized from the optical images. See the Supplementary Figure 13 artwork. This technique potentially provides a different route to the well-known resonator biosensors for in vitro disease diagnosis such as the detection of *Coxiella burnetii* in blood plasma from patients.⁴⁸” at the end of Conclusion.

2.3) The notation “perturbation” is in my opinion irritating and not well known to the microscope community. What the authors really detect are nanoscale objects.

We have replaced “perturbation” by “nanoscale objects” in the manuscript.

2.4) As the authors highlight (only once) throughout the manuscript, this is a near-field approach and this needs to be stressed more prominently, already starting in the abstract and introduction.

We have stressed this point more prominently both in the abstract and introduction. See “... to directly view $\lambda/31$ -scale (25-nm radius) objects in the nearfield region of nanowire-based sensors across ...” in the Abstract and “... and enable the detection of multiple nanoscale objects in the nearfield region of the nanowire assembly” in the third paragraph of Introduction.

2.5) The authors only show the detection of individual perturbations near the nanowires. Resolution is however about distinguishing multiple nearby features, objects or perturbations. Just observing individual perturbations such as individual bacterium or virus is also possible with conventional iSCAT.

We thank the reviewer for this insightful comment.

Indeed, as the reviewer correctly pointed out, our paper only shows the detection of individual perturbations near the nanowires. This is because our technique is not a super-resolution technique, but rather is the one that focuses on boosting the SNR of nanoscale objects (please see the description relating to Equations 1-3 in the manuscript). For observing individual perturbations, indeed, this is also possible with conventional iSCAT, and we have stated that in the manuscript for iSCAT. Please see “For the task of visualizing an isolated free-form object, our method outperforms brightfield and darkfield because of the non-resonant amplification from the nearby nanowire pair; however, iSCAT and ICPM outperform our method because they do not rely on an EC and thus have a larger signal strength. Moreover, by introducing advanced designs such as inserting a transmissive spatial mask placed into the back focal plane of a high numerical aperture microscope objective or coating a partially reflective metallic mask onto the center of a glass window,^{37,38} the contrast can be significantly enhanced when compared with iSCAT in its initial implementations. These enhanced iSCAT may even enable routine label-free imaging down to the single-molecule level” on Page 12.

We hope our clarification addresses the reviewer’s concern.

2.6) The comparison in figure 3 is somehow unfair. Dark-field/bright-field and iSCAT do not need the two parallel nanowires and would be able to detect single perturbations in the absence of the nanowires. Therefore, this might bring up the question why one should use the current approach (despite its simplicity).

We apologize for the confusion caused by the lack of detailed clarification of why we made figure 3.

Indeed, as the reviewer correctly pointed out, dark-field/bright-field and iSCAT do not need the two parallel nanowires when sensing individual nanoscale objects such as viruses and molecule clusters, which are well known to the biomedical community. However, they cannot be utilized in the case where nanoscale objects (for example, a line-connection defect or nanoscale particles) have to be symbiotic with parallel nanowires: for example, in semiconductor integrated circuit chips (as shown in the SEM image above, we can see a line-connection defect that connects two regular nanowires in a typical semiconductor IC chip. Adapted from Ref. 10 in the manuscript), there are particles or defects that need to be detected. Therefore, we have revised the related sentences in the hope that it can address the reviewer’s confusion. Please see “Based on the two-beam anti-symmetric excitation, we first use simulation to investigate the field enhancement of various fixed-form objects on a pair of nanowires (to mimic the defects in a semiconductor integrated circuit chips) and compare ...” at the beginning of the 2nd paragraph on Page 10.

2.7) Minor issues:

- Page 5, first sentence last paragraph of introduction: “We have” sounds like the authors refer to experiments published in the past.
- Page 8, bottom: Better to use “theoretically predicted” instead of “obtained” to highlight that these are not experimental data (I initially thought so).
- Page 10/11: The description of the comparison to dark-field is somehow missing (or did I miss this)?
- Page 11, bottom: Cite Fig S1-S4 for the quantification done e.g. on the contrast and gap dependence already here.
- Page 13, top: Is the polarization of the laser light important?
- Page 13, bottom: Was the best contrast determined visually or somehow quantitatively?
- Page 14, top: What is “best-focal”?

We thank the reviewer’s careful reading for our manuscript. Our revisions are listed below:

1). We revised the corresponding sentence. Please see “In this work, we demonstrated experimentally that we can visualize the shapes and positions...”

2). We have replaced “obtained” by “theoretically predicted” accordingly.

3). Thanks for pointing this out. We added “Darkfield microscopy has a higher contrast than that of brightfield and iSCAT, but is still smaller than that of the proposed framework; see the contrast curves on the right side of Fig. 3c” to Page 11.

4). We have added “Please see Supplementary Figures 1-4 in the Supplementary Information for the quantification done on the symmetry-induced contrast and gap dependence” on Page 11.

5). Yes. The strong bright modes (i.e. strong scattering signal) can be excited when the polarization of the illumination beam is parallel to the long axis of nanowires. Please see “The reason why we choose nanowires is that they can excite strong bright modes when the polarization of the illumination beam is parallel to the long axis of nanowires.^{42,43}” on Page 13.

6). It was determined visually. We have updated it to “...such that the EC with the best contrast can be found visually” on Page 14.

7). We mean “best focal plane.” We have revised it accordingly. Please see “Figure 4c shows the image of the double-nanowire structure in the best focal plane as the sample is...”

Sincerely,

Jinlong Zhu

Department of Electrical and Computer Engineering

REVIEWERS' COMMENTS Round Two:

Reviewer #1 (Remarks to the Author):

I have read the detailed and careful responses to the referee comments on the manuscript on "Visualizable Detection of Nanoscale Objects" by Jinlong Zhu et al and it is good to see that they have carefully considered comments by all referees and made extensive changes accordingly.

Although the responses don't fully convince me I find the revised manuscript interesting and consider it suitable for publication in Nature Communications in the current form (although I do regret the loss of the perturbations terminology, which I found helpful).

Even though I consider the manuscript publishable in its current form following the changes made I would still like to take this opportunity to comment on the detailed and thoughtful responses of the authors to my earlier comments as food for thought:

1) In my comment on the effect being claimed to be valid independent of the gap-size I was actually more concerned about how bigger gap-sizes interact with the dynamic range of the camera not the relatively small gaps of wavelength/40 or wavelength/256 simulated in the response by the authors. My thought experiment was trying to get across that going for gaps that are on the order of the wavelength or wavelength/2 the effect might get drowned in the much larger signal in the side-lobes coming from the excitation even though the center zero still exists.

This was in any case a very minor comment triggered by Figure 3 in the supplementary info and the response is convincing enough for what is included in my opinion.

2) I still remain to be convinced that this technique will ever be able to see small bio-molecules or viruses even on a silicon wafer, but that is probably for the community to figure out and explore based on the ideas presented in this work.

The aspect I was trying to get across in my comment is that the random roughness of the sample (in this case a silicon wafer) contributes to the background over the whole area in the diffraction limited spot corresponding to a pixel on the camera because all of this random roughness on the sample will break symmetry and will therefore yield an unavoidable background signal. As this background originates from the whole diffraction limited area my expectation is that the signal to background scales with the area and not with the RMS variation of the topography.

Based on that line of thinking and my understanding of the manuscript is that to see a 20 nm-scale biomolecule or virus I think that a wafer roughness below 0.2nm is required, not the 2nm stated simply because of this scaling of the background originating from sample roughness with the area not the height-variation!

Note that in reality this is probably made even worse due to the much smaller refractive index contrast of a bio-molecule in water (as drawn is supplementary figure 13) compared to the refractive index contrast of silicon vs water, which is not explicitly considered in the above discussion.

Reviewer #2 (Remarks to the Author):

In this revision and the reply to the reviewers, the authors have both improved the paper and provided a thorough and well stated response to all the questions and comments from each reviewer. I find the response letter most compelling. Thank you for your attention to details. I also find the supplemental information to be a critical part of this paper.

Reviewer #3 (Remarks to the Author):

The authors have very well replied to my previous concerns and have revised the manuscript well.

I am still highly supportive of publication.

The authors might want to consider to stress the advantage of their approach over conventional methods a little bit more, as they have written in the reply to my previous comments: "dark-field/bright-field and iSCAT do not need the two parallel nanowires when sensing individual nanoscale objects such as viruses and molecule clusters, which are well known to the biomedical community. However, they cannot be utilized in the case where nanoscale objects (for example, a line-connection defect or nanoscale particles) have to be symbiotic with parallel nanowires: for example, in semiconductor integrated circuit chips". But this is not mandatory.

UNIVERSITY OF ILLINOIS
AT URBANA - CHAMPAIGN

Micro and Nanotechnology Laboratory

Dr. Jinlong Zhu
2231 Micro and Nanotechnology Laboratory
208 North Wright Street
Urbana, IL 61801-2355

We thank the reviewers again for the insightful comments. Our responses to reviewers' individual points are shown below in blue and edits to the manuscript are shown in red.

REVIEWERS' COMMENTS:

Reviewer #1 (Remarks to the Author):

I have read the detailed and careful responses to the referee comments on the manuscript on "Visualizable Detection of Nanoscale Objects" by Jinlong Zhu et al and it is good to see that they have carefully considered comments by all referees and made extensive changes accordingly.

Although the responses don't fully convince me I find the revised manuscript interesting and consider it suitable for publication in Nature Communications in the current form (although I do regret the loss of the perturbations terminology, which I found helpful).

Agreed. We prefer use of the perturbations terminology and have added it back to the title and to the paper.

Even though I consider the manuscript publishable in its current form following the changes made I would still like to take this opportunity to comment on the detailed and thoughtful responses of the authors to my earlier comments as food for thought:

1) In my comment on the effect being claimed to be valid independent of the gap-size I was actually more concerned about how bigger gap-sizes interact with the dynamic range of the camera not the relatively small gaps of wavelength/40 or wavelength/256 simulated in the response by the authors. My thought experiment was trying to get across that going for gaps that are on the order of the wavelength or wavelength/2 the effect might get drowned in the much larger signal in the side-lobes coming from the excitation even though the center zero still exists.

This was in any case a very minor comment triggered by Figure 3 in the supplementary info and the response is convincing enough for what is included in my opinion.

Thanks for this clarifying note. We do present experimental data when the gap is $0.46 \times \text{wavelength}$ and $0.52 \times \text{wavelength}$ (see the vertical gaps in the last 2 columns of Fig. 4e) and when the gap is $0.96 \times \text{wavelength}$ (see the vertical gap in Fig. 5a). The side lobes do exist, but the two main peaks are significantly stronger and the two lines can be clearly visualized. We have thus added a sentence to the Discussion section:

"We have also visualized individual nanowires spaced by 0.10λ to 0.52λ (see Fig. 4c and e) and by 0.96λ (see Fig. 5a)."

2) I still remain to be convinced that this technique will ever be able to see small bio-molecules or viruses even on a silicon wafer, but that is probably for the community to figure out and explore based on the ideas presented in this work.

The aspect I was trying to get across in my comment is that the random roughness of the sample (in this case a silicon wafer) contributes to the background over the whole area in the diffraction limited spot corresponding to a pixel on the camera because all of this random roughness on the sample will break symmetry and will therefore yield an unavoidable background signal. As this background originates from the whole diffraction limited area my expectation is that the signal to background scales with the area and not with the RMS variation of the topography.

Based on that line of thinking and my understanding of the manuscript is that to see a 20 nm-scale biomolecule or virus I think that a wafer roughness below 0.2nm is required, not the 2nm stated simply because of this scaling of the background originating from sample roughness with the area not the height-variation!

Note that in reality this is probably made even worse due to the much smaller refractive index contrast of a bio-molecule in water (as drawn is supplementary figure 13) compared to the refractive index contrast of silicon vs water, which is not explicitly considered in the above discussion.

Agreed. This is something that the community will need to figure out and low index contrast will make this problem even harder. We have updated the sentence in the “Physical models of ECs and non-resonance amplification” section to read:

“This indicates that the background roughness (line edge and wafer surface) is small compared to the 20-nm or larger size of a typical bio-molecule such as a virus. However, because surface roughness is distributed over a large area and may present a strong index contrast, an optimized fabrication process to reduce surface roughness may be needed to visualize individual bio-molecules.”

We have also added a sentence “...by choosing nanowires with optimized geometry and proper refractive index (to make it comparable to that of the biological objects) and patterning...” at the end of Discussion to remind that in order to detect biological objects, a geometry optimization and an index match (for example, choose SiO₂ nanowires) may be required.

Lastly, we added Supplementary Note 6 for a more detailed discussion.

Reviewer #2 (Remarks to the Author):

In this revision and the reply to the reviewers, the authors have both improved the paper and provided a thorough and well stated response to all the questions and comments from each reviewer. I find the response letter most compelling. Thank you for your attention to details. I also find the supplemental information to be a critical part of this paper.

Thank you.

Reviewer #3 (Remarks to the Author):

The authors have very well replied to my previous concerns and have revised the manuscript well. I am still highly supportive of publication.

Thank you.

The authors might want to consider to stress the advantage of their approach over conventional methods a little bit more, as they have written in the reply to my previous comments: “dark-field/bright-field and iSCAT do not need the two parallel nanowires when sensing individual nanoscale objects such as viruses and molecule clusters, which are well known to the biomedical community. However, they cannot be utilized in the case where nanoscale objects (for example, a line-connection defect or nanoscale particles) have to be symbiotic with parallel nanowires: for example, in semiconductor integrated circuit chips”. But this is not mandatory.

The advantages were mentioned in various parts of the paper and they have now been consolidated into the discussion section for better clarity and emphasis.

Sincerely,

Jinlong Zhu

Department of Electrical and Computer Engineering
University of Illinois at Urbana-Champaign